# Distinct epigenetic programs regulate cardiac myocyte development and disease in the human heart in vivo

Ralf Gilsbach et al.#

Epigenetic mechanisms and transcription factor networks essential for differentiation of cardiac myocytes have been uncovered. However, reshaping of the epigenome of these terminally differentiated cells during fetal development, postnatal maturation, and in disease remains unknown. Here, we investigate the dynamics of the cardiac myocyte epigenome during development and in chronic heart failure. We find that prenatal development and postnatal maturation are characterized by a cooperation of active CpG methylation and histone marks at *cis*-regulatory and genic regions to shape the cardiac myocyte transcriptome. In contrast, pathological gene expression in terminal heart failure is accompanied by changes in active histone marks without major alterations in CpG methylation and repressive chromatin marks. Notably, *cis*-regulatory regions in cardiac myocytes are significantly enriched for cardiovascular disease-associated variants. This study uncovers distinct layers of epigenetic regulation not only during prenatal development and postnatal maturation but also in diseased human cardiac myocytes.

. Correspondence and requests for materials should be addressed to L.H. (email: lutz.hein@pharmakol.uni-freiburg.de). #A full list of authors and their affiliations appears at the end of the paper.

The heart is the first organ to develop during embryogenesis[1]. The general principles of cardiac development have been studied in great detail on a morphological and molecular basis. These studies have identified key signaling events and transcription factor networks that are involved in the specification and differentiation of cardiac myocytes (CMs)[1–6]. Many of these transitions involve changes of gene expression. This in turn is regulated by epigenetic processes including CpG methylation (mCpG) and histone modifications. However, the detailed epigenetic processes involved in maturation from fetal to adult CMs and in cardiac disease leading to terminal heart failure have not been fully uncovered, yet.

Epigenetic studies in human hearts have identified altered mCpG in chronic heart failure[7,8]. Owing to technical restrictions, these studies were performed in heart tissue and therefore the affected cell type(s) could not be identified. Epigenetic mechanisms are highly cell-type-specific requiring cell separation techniques to determine epigenomic features in a specific cell type, especially when keeping in mind that the cellular composition of the human heart is highly dynamic. Previous studies in mice have used enzymatic dissociation by the Langendorff technique[9,10] or additional purification of dissociated cardiac cells by flow cytometry[11,12] to study cell-type-specific features of the CM transcriptome and epigenome, respectively. Purification of CM nuclei by fluorescence-assisted sorting has led to the identification of cell-type-specific mCpG and histone modification signatures in CMs during mouse heart development and maturation[11]. Based on this method[13,14], we used a nuclear staining strategy to isolate CM nuclei from intact prenatal and postnatal human heart tissue and subjected these nuclei to comprehensive analysis of the epigenome during prenatal development, postnatal maturation, and in heart failure.

Here we describe the human CM epigenome during prenatal development and postnatal maturation of the heart from infant to adult age and in terminal failure. We find that during normal lifespan of CMs gene regulation is mainly orchestrated by dynamic mCpG and canonical histone marks at distal regulatory and genic regions. In contrast to previous findings in heart tissue, expression of the pathological gene program in heart failure was not accompanied by changes in the CM DNA methylome but by active histone marks. In addition, our study provides a functional map of the non-coding genome of human CMs throughout life. Linking this functional annotation with known genetic polymorphisms revealed the presence of cardiovascular disease-associated polymorphisms in active CM enhancers.

## Results

**Flow cytometry and sorting of human CM nuclei.** To determine the dynamics of the CM epigenome during prenatal development and postnatal maturation as well as in disease, cardiac left ventricular (LV) tissue was analyzed at three stages (fetal, 16–23 weeks of pregnancy; infant, 1–12 months of age; adult, 46–60 years) and at one disease condition of end-stage heart failure (56–63 years, LV ejection fraction $19 \pm 2\%$, Supplementary Data 1). Fluorescence-activated sorting of cardiac nuclei was applied to ensure CM specificity (Fig. 1a). Antibodies raised against pericentriolar material 1 (PCM1) or signal regulatory protein alpha (SIRPA) were used to stain postnatal CM nuclei and fetal CMs, respectively[11,14,15]. These markers are highly specific for postnatal and fetal CMs, respectively[11,14,15]. In addition, we identified and validated phospholamban (PLN) as a specific marker for prenatal and postnatal CM nuclei. In the heart, only CMs show high abundance and strong nuclear localization of PLN[16,17]. We confirmed the validity of nuclear PLN using dual labeling of CM nuclei with PCM1 by flow cytometry

(Fig. 1b, Supplementary Figs. 1 and 2). Fetal and infantile hearts contained a higher percentage of CM nuclei than adult hearts (Fig. 1c). Notably, adult CM nuclei showed a higher degree of ploidy than fetal and infantile CM nuclei (Fig. 1d). While fetal and infantile hearts mostly contained diploid CM nuclei, the percentage of tetraploid nuclei increased significantly until adulthood (Fig. 1d). Adult failing myocytes showed even a higher proportion of polyploid nuclei (Fig. 1d).

CMs were isolated with high purity (≥98%) from LV tissue (Fig. 1e, Supplementary Fig. 2). Purified nuclei were processed to generate high-coverage DNA methylomes (whole-genome bisulfite sequencing (WGBS)), 5-hydroxymethylomes (5-hydroxy-methyl-cytosine sequencing (5hmC-seq)), profiles of seven histone marks (chromatin immunoprecipitation sequencing (ChIP-seq)) and nuclear gene expression (RNA sequencing (RNA-seq)). All data sets were derived from independent biological replicates (Supplementary Data 2). For nine patients, at least six histone marks, mCpG, and nuclear RNA expression were generated from the same samples (Supplementary Data 2). As 5hmC-seq required higher amount of input DNA, different biological replicates were pooled (Supplementary Data 2). In total, we generated >3 billion mapped ChIP-seq and 60 million 5hmC-seq reads as well as 1 billion mapped RNA-seq fragments. The cumulative CpG coverage resulting from WGBS was 135-fold (Supplementary Data 2). Data generated from independent biological replicates yielded highly correlating values for ChIP-seq (Supplementary Fig. 3), RNA-seq (Supplementary Fig. 4a), and for mCpG (Supplementary Fig. 5). Sequencing data further confirmed the validity of the fluorescence-activated cell sorting (FACS) strategy for CM nuclei. CpG methylation data from SIRPA-sorted cells correlated highly with data obtained from PCM1- and PLN-sorted nuclei (Supplementary Fig. 5, correlation coefficient ≥0.95). Sequencing of nuclear RNA showed expression of CM genes in PLN- and PCM1-positive nuclei and no expression of non-CM genes (Supplementary Fig. 6a). RNA profiles of PCM1- and PLN-sorted nuclei were highly correlated ($R^2 = 0.96$, Supplementary Fig. 6b).

Representative traces show a genomic region containing troponin T2 (*TNNT2*) and troponin I1 (*TNNI1*, Fig. 1f). *TNNT2* is highly expressed in CMs from fetal to adult stages and shows sequential loss of genic mCpG and a promoter enrichment of H3K27ac, H3K9ac, and H3K4me3 and genic enrichment of H3K36me3 (Fig. 1f). The fetally expressed *TNNI1* gene was silenced postnatally, coinciding with genic de novo mCpG loss and loss of active histone marks H3K27ac, H3K9ac, H3K4me3, and H3K36me3 (Fig. 1f). Several regions with distal regulatory domain signatures, including low mCpG and enrichment of H3K4me1, were identified. While most of these regions seem to be relatively stable at this genomic locus, dynamic establishment of a distal regulatory signature occurred during CM development in an intronic region of the plakophilin (*PKP1*) gene (Fig. 1f).

Next, epigenetic signatures of heart tissue[18–20] and CM were compared to get insight into cell-type specificity (Supplementary Fig. 7). Loci containing the CM-specific myosin heavy chain α and β genes as well as the fibroblast gene biglycan[21] or the endothelial gene VE-cadherin revealed distinct differences in mCpG and chromatin state between tissues and CMs (Supplementary Fig. 7b–d). Genes that were hypomethylated in CMs vs. heart tissue (Supplementary Fig. 7e, group 1) were involved in cardiac muscle function (Supplementary Fig. 7f, group 1). In contrast, genes with higher levels of mCpG in myocytes vs. heart tissue represented developmental processes of the cardiovascular system, vasculature, and connective tissue (Supplementary Fig. 7e, f, group 2). Elevated mCpG was also observed in promoter regions flanking the transcription start site (TSS; Supplementary Fig. 7g).

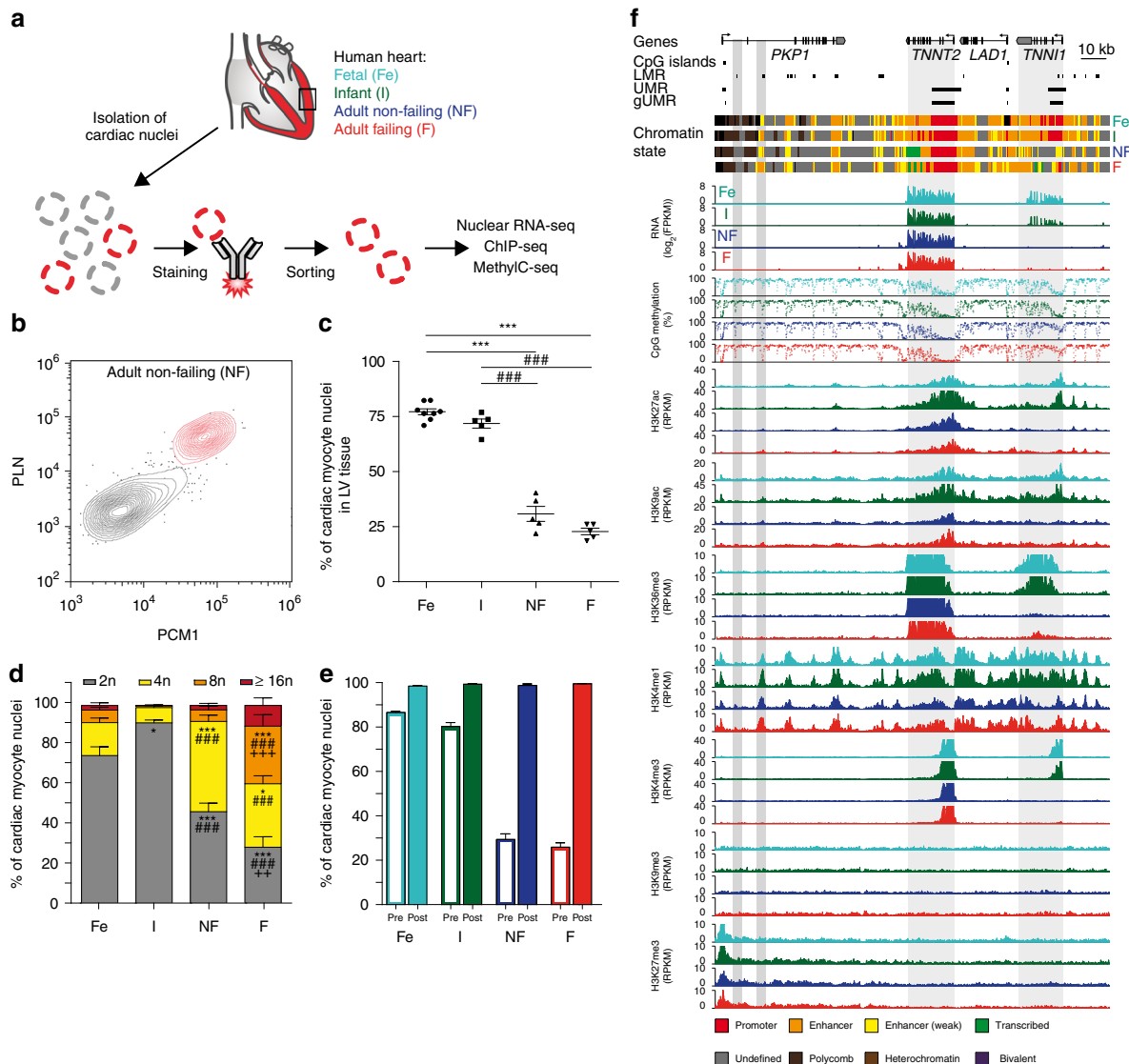

**Fig. 1** Sorting and analysis of cardiac myocyte nuclei and representative epigenetic map. **a** Workflow for the isolation of human cardiac myocyte nuclei for epigenetic and transcriptomic analysis. **b** FACS analysis of nuclei isolated from adult non-failing left ventricular tissue (LV). Nuclei were stained with anti-pericentriolar material 1 (PCM1) and anti-phospholamban (PLN) antibodies to identify cardiac myocyte nuclei (red). **c** Proportion of cardiac myocyte nuclei in fetal (Fe, $n = 8$), infantile (I, $n = 5$), adult non-failing (NF, $n = 5$), and adult failing (F, $n = 5$) LV tissue (mean ± SEM). **d** Distribution of cardiac myocyte ploidy in fetal ($n = 8$), infantile ($n = 5$), non-failing ($n = 5$), and failing ($n = 5$) left ventricles (mean ± SEM). **e** Percentage of cardiac myocyte nuclei in LV tissue before sorting (pre, open columns) and cardiac myocyte nuclei purity after FACS sorting (post, $n = 3$, mean ± SEM). **f** Original traces of RNA expression, mCpG, and histone marks of the troponin I type 1 (*TNNI1*) and troponin T type 2 (*TNNT2*) gene region. CpG islands, low methylated regions (LMR), unmethylated regions (UMR), genic unmethylated regions (gUMR), and the chromatin state are annotated. Gray areas highlight differentially CpG-methylated regions and genes. Shown are data from $n$ biological replicates: mCpG, $n = 3$–5; H3K27ac, H3K9ac, H3K36me3, H3K4me1, H3K4me3, H3K27me3, and H3K9me3, $n = 3$; RNA, $n = 3$–4; * vs. Fe, $p < 0.05$; ++ vs. NF, $p < 0.01$; *** vs. Fe, ### vs. I, +++ vs. NF, $p < 0.001$ by ANOVA

Systematic analysis of CM and VISTA heart enhancers[22] revealed an overlap of 49% (Supplementary Fig. 7h). VISTA enhancers not present in CMs were adjacent to genes involved in artery morphogenesis and collagen fibril organization while CM-specific ones were adjacent to genes related to cardiac muscle contraction and morphogenesis (Supplementary Fig. 7i, j). Narrowing the list of VISTA heart enhancers down to those with confirmed function in mouse embryonic hearts in vivo[22] showed a 90% overlap with CM enhancers (Supplementary Fig. 7k).

**mCpG-guided annotation of genomic elements**. Genomic regions, including distal regulatory domains and genic regions, show distinct mCpG patterns. Therefore, we segmented the

genome into partially methylated, unmethylated, and low methylated regions (PMR, UMR, and LMR, respectively)[23,24]. We identified large PMRs with highly disordered mCpG covering 43% of the genome (Supplementary Figs. 8 and 9a). PMRs spanning large genomic domains have been observed in other somatic cell types[25] too but were not detectable in mCpG data obtained from human heart tissues[18]. In agreement with previous studies[26], most CM PMRs (Supplementary Fig. 10a) showed low levels of active histone marks (Supplementary Fig. 10b–d) and are associated with a silent chromatin state (Supplementary Fig. 11a, b).

Further segmentation of mCpG data resulted in UMRs and LMRs, which covered 3% of the genome each (Supplementary Fig. 9a). LMRs (Supplementary Fig. 10e–h) showed enrichment

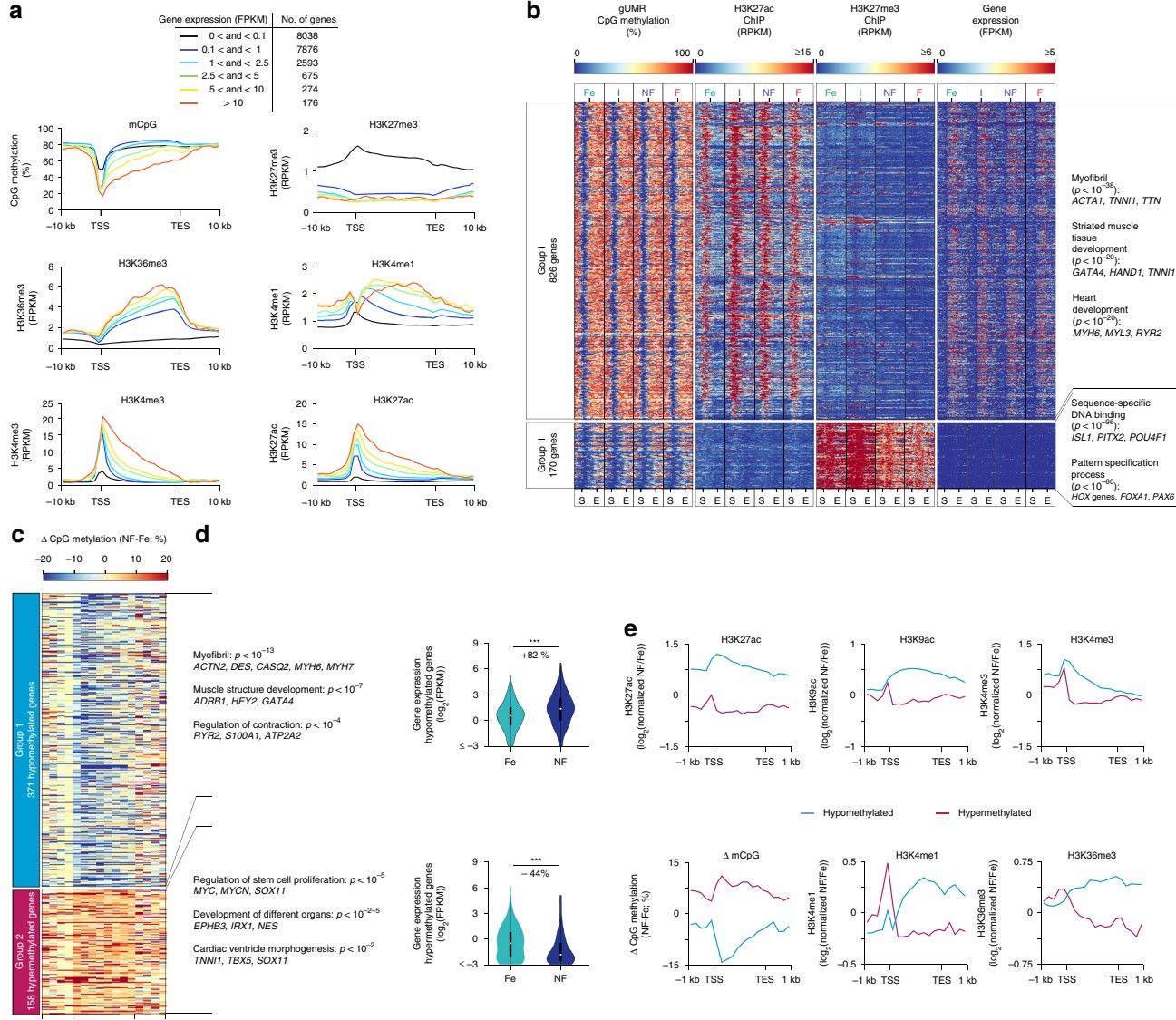

**Fig. 2** Genic CpG demethylation of cardiac myocyte genes. **a** Genes were grouped according to their mean expression level (FPKM) in adult non-failing cardiac myocyte nuclei. Numbers of genes in each expression group are listed in the table. Average plots for mCpG and levels of H3K4me3, H3K4me1, H3K36me3, H3K27ac, and H3K27me3 are represented from TSS (transcription start site) to TES (transcription end site) including 10 kb flanking regions for each group. **b** Characterization of genes with pronounced genic demethylation. Criteria for these genes were gUMR length ≥5 kb and/or gUMR overlapping with ≥25% of a gene. Heatmaps of mean gUMR mCpG, enrichment of genic H3K27ac and H3K27me3, and gene expression. Genes were clustered into two groups according to developmental alteration of mCpG and presence of H3K27me3. Gene ontology analysis of groups I and II shows most highly enriched GO terms, representative genes, and Bonferroni corrected $p$-values. **c** Analysis of genes with gUMR-DMRs between fetal and adult non-failing cardiac moycytes. **d** Gene ontology analysis of genes with differential gUMR methylation between fetal and adult non-failing cardiac myocytes. The list shows most highly enriched GO terms, representative genes, and Bonferroni corrected $p$-values. Gene expression of genes with hypomethylated (upper graph) or hypermethylated gUMRs (lower graph) in adult non-failing vs. fetal cardiomyocytes is displayed as violin plots with inserted box and whisker plots. ***$p < 0.001$ by ANOVA. **e** Changes in active histone marks (upper graphs), mCpG, 5hmC levels and H3K27me3 (lower graphs) in hypomethylated (A, group 1) or hypermethylated (A, group 2) gene bodies in adult non-failing *vs.* fetal cardiac myocytes. Figures show data from $n$ biological replicates: mCpG, $n = 3–5$; H3K27ac, H3K9ac, H3K36me3, H3K4me1, H3K4me3 and H3K27me3, $n = 3$; RNA, $n = 3–4$

for H3K4me1 (Supplementary Fig. 10g), depletion of H3K4me3 (Supplementary Fig. 10h), and were associated with an enhancer-associated chromatin state (Supplementary Fig. 11a, b). Overall, 73–78% of LMRs showed an enhancer chromatin signature (Supplementary Fig. 11d) and 55% overlap with strong enhancers (Supplementary Fig. 11e). Genes adjacent to LMRs with enhancer chromatin signature were significantly higher expressed as genes neighboring LMRs with non-enhancer chromatin state (Supplementary Fig. 11e). LMRs were strongly enriched for transcription factor motifs (Supplementary Fig. 10m) and were predominantly

found in intronic and intergenic regions (Supplementary Fig. 10n). The five most significantly enriched motifs contained binding sites for the CCCTC-binding factor and transcription factors of the myocyte enhancer factor-2 (MEF2), GATA, TGGCA-binding protein (CTF/NF1), and T-box families (Supplementary Fig. 10m).

Compared to LMRs, UMRs were larger in size (Supplementary Fig. 10i vs. e), showed enrichment of the promoter mark H3K4me3 (Supplementary Fig. 10l), and predominantly spanned genic regions and annotated CpG islands (Supplementary

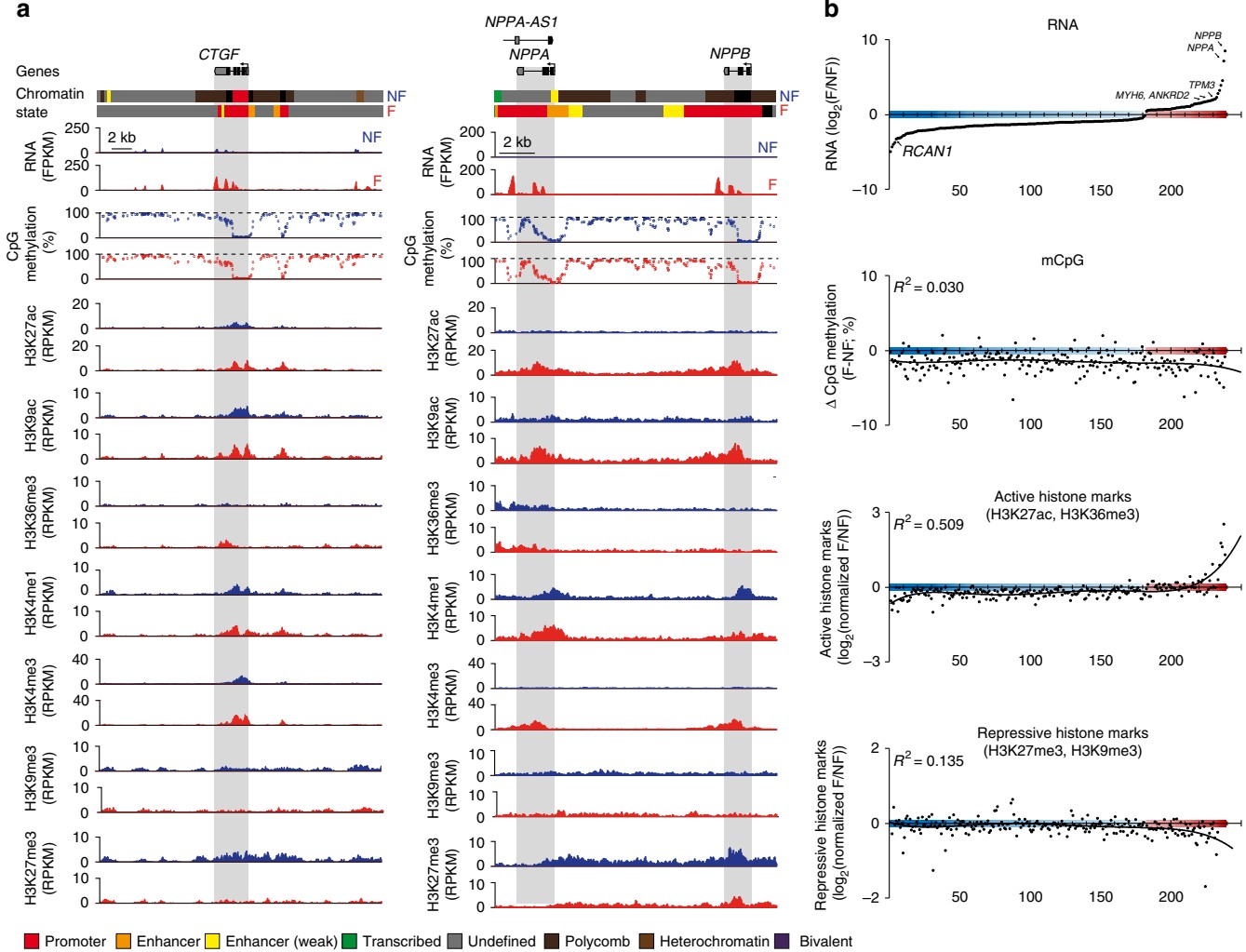

**Fig. 3** mCpG and histone profiles of genes that are differentially expressed in failing cardiac myocytes. **a** Original traces of two loci containing connective tissue growth factor (*CTGF*) and the two natriuretic peptide A and B genes (*NPPA* and *NPPB*). Traces include gene expression, mCpG, histone modifications, and the annotated chromatin state. Genic regions are highlighted in gray. **b** Differentially expressed genes in cardiac myocyte nuclei of non-failing compared to failing hearts were ranked according to expression changes (top panel). The relative enrichment of active (H3K27ac, H3K36me3) and repressive (H3K9me3, H3K27me3) histone marks together with changes in mCpG are shown for the ranked genes. Resulting $R^2$-values of a non-linear regression (black line) are indicated in each graph. Representative genes of the fetal gene program are labeled in bold letters. Figures show data from $n$ biological replicates: mCpG, $n = 5$; H3K27ac, H3K9ac, H3K36me3, H3K4me1, H3K4me3, H3K27me3, and H3K9me3, $n = 3$

Fig. 10n, o). To specifically analyze the characteristics of genes with low methylated gene bodies, we annotated genic UMRs (gUMR). They span genic parts overlapping with TSSs. To eliminate a potential bias, regions overlapping with CpG islands were excluded from gUMR annotation (Fig. 1f, Supplementary Fig. 10o).

**gUMRs in CMs**. Next, we analyzed the correlation of gene expression with inheritable marks (Fig. 2a, Supplementary Fig. 12). Ranking of genes according to their gene expression level revealed an inverse correlation of mCpG and gene expression around the TSS (Fig. 2a, Supplementary Fig. 12). Analysis of histone modifications showed a positive correlation of active histone marks (H3K27ac, H3K9ac, H3K36me3, H3K4me1, H3K4me3) with gene expression, while genes with very low transcriptional activity (<1 fragments per kilobase per million fragments mapped (FPKM)) were decorated with the inactive marks H3K27me3 and H3K9me3 (Fig. 2a, Supplementary Fig. 12). Comparing genes expressed (≥1 FPKM) in at least one of the assessed stages (6515 genes, Fig. 2a) revealed that 45% were

expressed independent of age or disease. The largest number of genes with stage-specific expression was detected in fetal CMs (>16%, Supplementary Fig. 4b). These genes were significantly associated with cell cycle and chromatin organization (Supplementary Fig. 4b).

The marked CpG demethylation of genic regions encouraged us to characterize genes with extended genic CpG demethylation. We identified 996 genes with a gUMR covering >25% or at least 5 kb of the gene (Fig. 2b, Supplementary Data 3). The majority of these genes were marked with active histone modifications including H3K27ac and were depleted for the polycomb mark H3K27me3 (Fig. 2b, Supplementary Fig. 13, group I). They showed a characteristic mCpG pattern with regions of low mCpG overlapping with the TSS and progressing toward the 3′ end of the gene (Fig. 2b, Supplementary Fig. 13b, group I). These genes showed high transcriptional activity (Fig. 2b, Supplementary Fig. 13a, group I) and include myofibril proteins or are known to be implicated in muscle structure development and muscle structure system processes (Fig. 2b, group I, Supplementary Data 3).

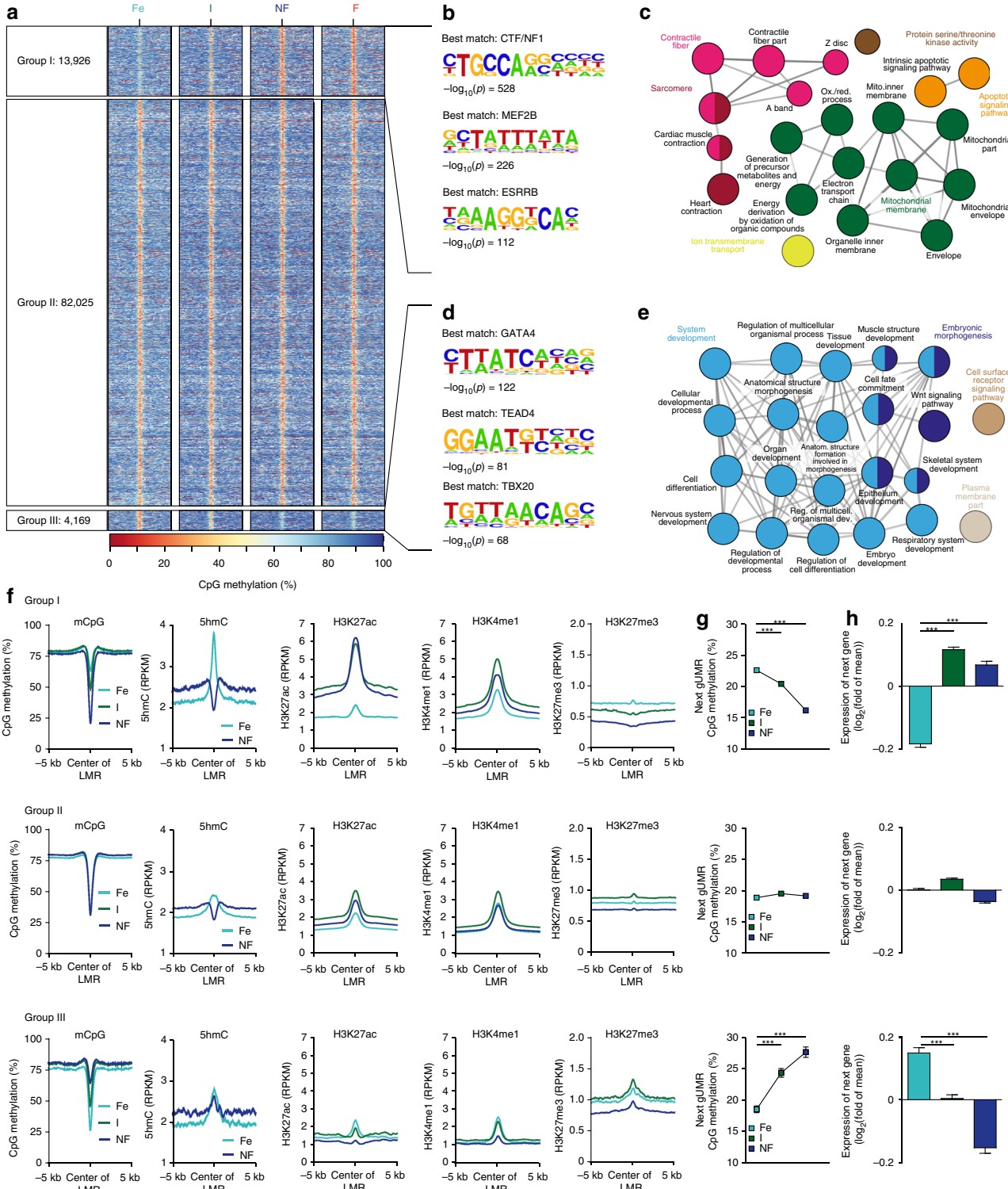

**Fig. 4** mCpG of low methylated regions (LMR) during development and maturation of cardiac myocytes. **a** Heatmap of mCpG within LMRs. LMRs were clustered into three groups according to differential mCpG in fetal vs. non-failing adult cardiac myocytes Shown are 10 kb windows around the LMR center. **a** Group I contains DM-LMRs with a loss, group III by gain, and group II by no significant change in mCpG. **b**–**e** Transcription factor enrichment within LMRs (**b**, **d**) and Gene Ontology analysis of adjacent genes (**c**, **e**). The depicted GO terms were significantly enriched with Bonferroni step down corrected *p*-values < 10⁻². Given are HOMER transcription factor enrichment *p*-values. **f** Profiles of mean mCpG and enrichment of histone modifications (H3K4me1, H3K9ac, H3K27ac, H3K27me3) at LMRs for groups I-III. **g**, **h** Next gUMR mCpG level (**g**) and relative gene expression of adjacent genes (**h**) (mean ± SEM). ***$p < 0.001$ by ANOVA. Figures show data from *n* biological replicates: mCpG, $n = 3$–5; H3K27ac, H3K9ac, H3K36me3, H3K4me1, H3K4me3, and H3K27me3, $n = 3$; 5hmC, pooled analysis from 2–5 biological samples, RNA, $n = 3$–4

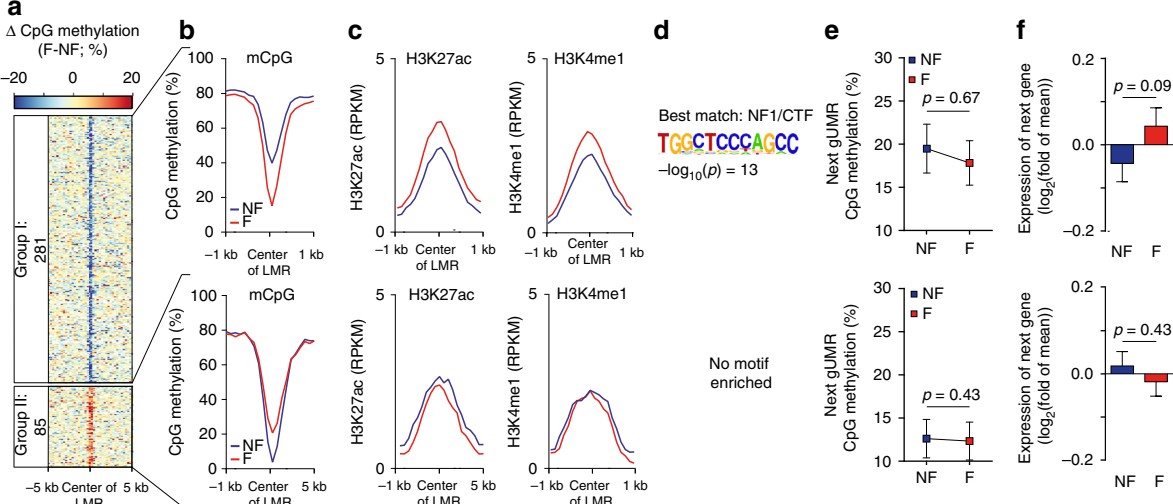

**Fig. 5** Characterization of LMRs and mCpG in heart failure. **a** Analysis of LMRs with differential mCpG between adult non-failing and failing cardiac myocytes. **b**, **c** Profiles of mCpG (**b**) and histone marks (**c**) at DM-LMRs that are hypomethylated or hypermethylated between failing and non-failing cardiac myocytes. **d** Significantly enriched transcription factor (TF) binding motifs in hypomethylated DM-LMRs. Hypermethylated DM-LMRs (group II) did not show significant enrichment for TF motifs. Given are HOMER transcription factor enrichment $p$-values. **e**, **f** Next gUMR mCpG level (**e**) and relative gene expression of adjacent genes (**f**) (mean ± SEM). Figures show data from $n$ biological replicates: mCpG, $n = 3$–5; H3K27ac, H3K4me1, $n = 3$; RNA, $n = 3$–4

The remaining subset of genes (Fig. 2b, group II) showed the typical signature of inactive chromatin, including enrichment of H3K27me3 (Fig. 2b, Supplementary Fig. 13i, j, group II) and depletion of active histone marks (Fig. 2b, Supplementary Fig. 13d–f, h, group II). This group of genes showed stable low mCpG and no significant transcriptional activity (Fig. 2b, Supplementary Fig. 13a, group II). These genes were associated with DNA binding and pattern specification (Fig. 2b, group II, Supplementary Data 3).

**Dynamic gUMR mCpG in CMs**. To characterize the dynamics of genic mCpG during cardiac maturation from fetal until adult stages and in disease, all 14,688 genes harboring a gUMR were analyzed. From fetal life until adulthood, 529 genes exhibited differential gUMR mCpG (DM-gUMRs) (Fig. 2c). This affected >16,000 CpGs (Supplementary Fig. 9e). Loss of mCpG in DM-gUMRs was associated with increased expression of genes essential for myofibril and sarcomere structures as well as regulation of contraction (Fig. 2d, group 1). In contrast, a developmental increase of mCpG in DM-gUMRs was linked to decreased gene expression and affected primarily developmental genes (Fig. 2d, group 2). Changes in genic mCpG were accompanied by concordant changes of histone marks (Fig. 2e). Demethylated genic regions during maturation gained the active histone marks H3K27ac, H3K9ac, H3K36me3, and H3K4me3 (Fig. 2e). In contrast, hypermethylated regions showed a loss of these marks (Fig. 2e). This illustrates the tight link of genic mCpG and histone marks with gene expression during myocyte development and postnatal maturation.

**Development vs. maturation of gUMR mCpG**. We separately analyzed gUMR mCpG during prenatal development and postnatal maturation (Supplementary Fig. 14). We found 29 gUMRs that were hypomethylated and 12 gUMRs that were hypermethylated from fetal to infant stage (Supplementary Fig. 14a, group I, II). Postnatally, 188 gUMRs were further demethylated and 34 gUMRs were hypermethylated (Supplementary Fig. 14a, groups III, IV). In all, 59% of genes that were differentially methylated prenatally showed continuing methylation changes

postnatally. No genes with opposing mCpG changes during prenatal vs. postnatal life were identified. We further performed a genome-wide principal component analysis of genic mCpG. This resulted in a trajectory from prenatal development to postnatal maturation (Supplementary Fig. 15a). Comparable trajectories were observed for gene expression and histone modifications (Supplementary Fig. 15b–i). Thus mCpG developed continuously from prenatal to postnatal life.

**gUMR mCpG is stable in chronic heart failure**. Remarkably, only 6 out of the 14,688 gUMR genes were significantly altered (DM-gUMRs) in adult failing vs. non-failing CMs (Supplementary Fig. 16). These differences were not accompanied by consistent alterations in gene expression (Supplementary Fig. 16).

**Epigenetic signature of disease-associated gene expression**. Since we could not identify a clear link between DM-gUMRs and RNA expression, we analyzed epigenetic signatures of differentially expressed genes in failing vs. non-failing adult CM nuclei (Fig. 3). For several pathologically relevant genes, including connective tissue growth factor (*CTGF*) and natriuretic peptides A and B (*NPPA*, *NPPB*), we identified concordant changes in active histone marks (H3K27ac, H3K4me3, H3K9ac, and H3K36me3) and gene expression in failing CMs (Fig. 3a). However, these genes did not show overt alterations in promoter or gene body mCpG (Fig. 3a).

To assess this on a genome-wide scale, we ranked all differentially expressed genes in failing CM nuclei according to the magnitude of disease-associated changes in expression (Fig. 3b). Changes in gene expression were accompanied by concordant alterations in active histone marks. The best predictive marks for gene expression were H3K27ac and H3K36me3 (Supplementary Fig. 17a, c). In combination, these marks explained 50% of the gene expression rank (Fig. 3b, $R^2 = 0.509$). For other histone modifications as well as for mCpG in gUMRs (Supplementary Fig. 17) and genic regions (Fig. 3b), we did not observe $R^2$ values exceeding 0.14. This suggests that disease-associated changes in gene expression primarily involve

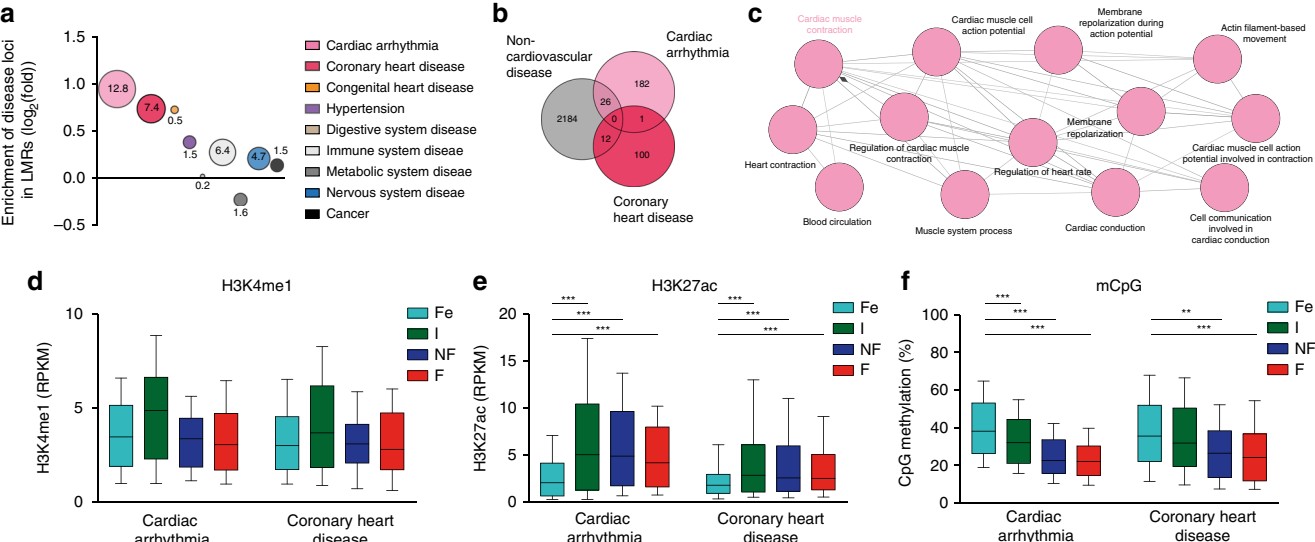

**Fig. 6** Polymorphisms associated with heart disease are enriched in cardiac myocyte LMRs. **a** Disease-associated polymorphisms overlap with cardiac myocyte LMRs. Single-nucleotide polymorphisms (SNPs) were extracted from GWAS studies and proxy SNPs in linkage disequilibrium were identified. Enrichment of SNPs in cardiac myocyte LMRs as compared to randomly sampled genomic regions was calculated for different disease traits. Shown are fold-enrichments. Circular areas reflect the respective level of significance. Numbers indicate $-\log_{10}$(Chi-squared $p$-values). **b** Venn diagram of LMR-containing SNPs, which have been linked with cardiac arrhythmia, coronary heart disease, or non-cardiovascular disease SNPs (digestive, immune, metabolic, and nervous system disease as well as cancer). **c** GREAT analysis of LMRs containing cardiac arrhythmia SNPs. Shown are enriched biological processes with FDR-corrected $p$-values < $10^{-8}$. **d–f** Enrichment of histone marks and mCpG in cardiac arrhythmia or coronary heart disease loci overlapping with cardiac myocyte LMRs. Shown are box plots with whiskers (5–95th percentile). Figures show data from $n$ biological replicates: mCpG, $n = 3$–5; H3K27ac, H3K4me1, $n = 3$; RNA, $n = 3$–4. ** vs. Fe, $p<0.01$; *** vs. Fe, $p < 0.001$ by ANOVA

remodeling of distinct active histone modifications, while especially repressive marks remain relatively stable.

**Reconfiguration of distal regulatory domain mCpG.** Segmentation of mCpG data identified >100,000 LMRs with distal regulatory properties (Fig. 4a, Supplementary Figs. 9c, 10e–h, and 11b). Of these regions, 18% (Fig. 4a, groups I+III, Supplementary Data 4) were differentially methylated (DM-LMR) between fetal and adult stages. The main fraction (77%) of DM-LMRs showed a developmental loss of mCpG (Fig. 4, group I) and was strongly enriched for transcription factor-binding motifs for CTF/NF1 and MEF2 (Fig. 4b). These DM-LMRs were located next to genes essential for cardiac contraction and energy supply (Fig. 4c). In contrast, DM-LMRs with increasing mCpG in adult vs. fetal CMs (Fig. 4a, group III) were enriched for DNA sequences similar to known motifs of GATA, TEAD, and T-box transcription factors (Fig. 4d). These DM-LMRs were adjacent to genes involved in general and muscle-specific early developmental and differentiation processes (Fig. 4e).

Loss of mCpG in DM-LMRs was accompanied by an increase of the active histone marks H3K27ac and H3K4me1 and a concordant decrease of H3K27me3 enrichment (Fig. 4f, group I). These DM-LMRs were associated with neighboring genes that lost gUMR methylation and showed increased expression (Fig. 4g, h, group I). CpG demethylation of these DM-LMRs was preceded by establishment of 5hmC at the center of the nascent LMRs indicating an active TET-dependent demethylation (Fig. 4f, group I). LMRs with stable mCpG showed a relatively constant level of histone marks, gene expression of neighboring genes, and their gUMR mCpG (Fig. 4f–h, groups II). LMR-diffentially methylated regions (DMRs) with increased mCpG from the fetal to the adult stage contained a primarily inactive histone signature as illustrated by high H3K27me3 and low H3K4me1 as well as H3K27ac signals (Fig. 4f, group III). Neighboring genes showed

an increase in genic mCpG and a decrease in gene expression (Fig. 4g, h, group III). This demonstrates an interplay of mCpG and histone marks at regulatory domains and their influence on transcriptional activity during development and maturation of CMs.

**Development vs. maturation of mCpG of LMRs.** Comparing LMRs during prenatal development and postnatal maturation indicated that loss of mCpG prevails in postnatal CMs (7160 hypomethylated vs. 693 hypermethylated DM-LMRs, Supplementary Fig. 14b). During development, the number of DM-LMRs with loss or gain of mCpG were comparable (Supplementary Fig. 14b). We found a similar set of transcription factor motifs to be enriched in DM-LMRs during the prenatal and postnatal periods (Supplementary Fig. 14c). Comparing development and maturation, only 0.36% (35) of DM-LMRs showed opposing mCpG (not shown).

**LMRs in heart failure.** Next, we analyzed DM-LMRs of adult failing vs. non-failing CM nuclei (Fig. 5). Heart failure affected the mCpG levels of 3647 CpGs within 366 DM-LMRs (Fig. 5a, Supplementary Fig. 9c). The extent of mCpG changes was >45-fold smaller in disease as compared to development (176,630 CpGs within 18,095 DM-LMRs) (Supplementary Fig. 9c).

DM-LMRs with a disease-associated loss of mCpG showed a tendency for higher levels of H3K27ac and H3K4me1 (Fig. 5b, c, group I). These DM-LMRs were enriched for sequences similar to the transcription factor motif of CTF/NF1 (Fig. 5d, group I). Genes adjacent to these DM-LMRs showed stable genic mCpG (Fig. 5e, group I) and gene expression (Fig. 5f, group I). GREAT analysis[27] of these genes did not reveal an association with cardiac gene programs. We identified 85 DM-LMRs with increasing mCpG in adult failing as compared with non-failing CMs (Fig. 5a, b, group II). These DM-LMRs did not show changes of active

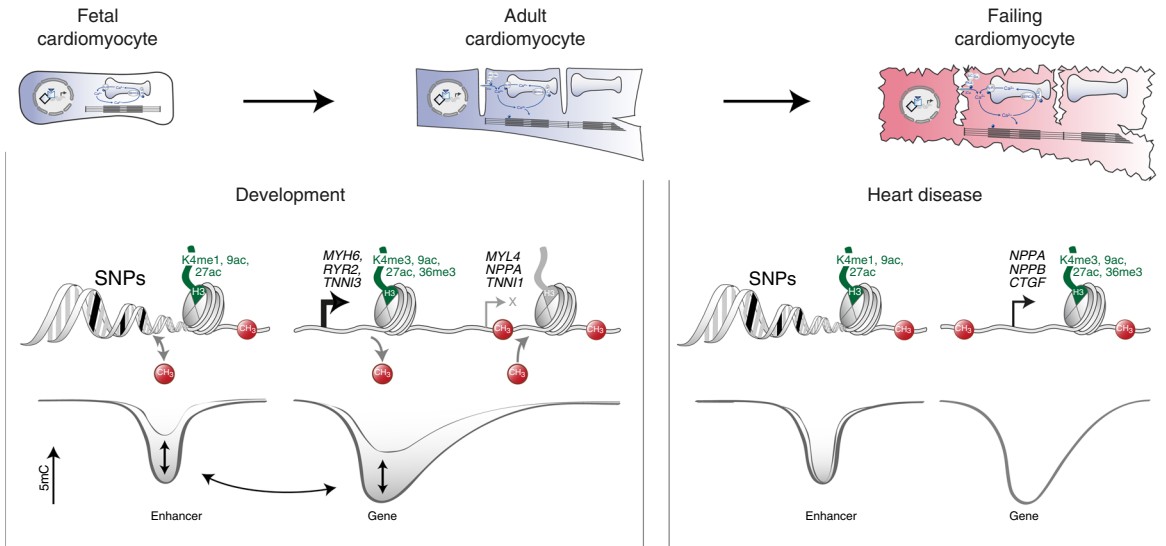

**Fig. 7** Dynamics of mCpG and histone modifications in gene bodies and enhancer regions of cardiac myocyte genes during heart development and in chronic heart failure. During cardiac myocyte development, mCpG of enhancers (LMR low methylated region, Fig. 4) and genic regions (gUMR genic unmethylated region, Fig. 2) and canonical histone marks cooperate to induce (*MYH6, RYR2, TNNI3*) or repress (*MYL4, NPPA, TNNI1*) cardiac myocyte genes (left panel, Fig. 1f). In failing cardiac myocytes (right panel), induction of disease-associated genes is accompanied by active histone marks without changes in gene body mCpG (Fig. 3b). Some LMRs showed small changes in mCpG in heart failure (Fig. 5a). Low methylated regions with enhancer signature were significantly enriched for single-nucleotide polymorphisms (SNP), which have been linked with cardiac disease (Fig. 6)

histone marks (Fig. 5c, group II), gUMR mCpG, or gene expression of adjacent genes (Fig. 5e, f). This indicates that disease-associated alterations of mCpG are rare and not directly linked to alterations of other inheritable marks or gene expression.

**Enrichment of genetic variants in LMRs.** Cardiac *cis*-regulatory regions contain disease-associated single-nucleotide polymorphisms (SNPs)[28]. To test whether genetic variations map to LMRs in CMs, we analyzed SNPs linked with different disease traits. SNPs associated with cardiac arrhythmia or coronary heart disease were most significantly enriched in CM LMRs (Fig. 6a). SNPs associated with congenital heart disease were also enriched in LMRs, but this association did not reach statistical significance (Fig. 6a). SNPs associated with diseases in other organ systems showed smaller or no enrichment in CM LMRs (Fig. 6a). LMRs associated with disease SNPs showed a very low overlap between cardiac arrhythmia, coronary heart disease, and non-cardiovascular disease, respectively (Fig. 6b). GREAT analysis identified that LMR regions associated with cardiac arrhythmia were enriched for genes involved in cardiac excitation and conduction or cardiac contraction (Fig. 6c). Disease-associated LMRs showed characteristics of *cis*-regulatory regions, including enrichment of H3K4me1 and H3K27ac (Fig. 6d, e). These regions showed increasing H3K27ac levels (Fig. 6e) and a significant loss of mCpG (Fig. 6f) from fetal to adult stages. This indicates that genetic polymorphisms that are linked with cardiac disease traits are enriched in *cis*-regulatory regions of CMs.

**Discussion**

Here we present epigenome maps of human CMs during prenatal development, postnatal maturation, and in chronic heart failure. Our data reveal a highly dynamic interplay between mCpG and histone modification to shape the CM transcriptome during development and maturation (Fig. 7). In chronic heart failure, mCpG of genic and *cis*-regulatory regions was remarkably stable. Pathological gene expression was accompanied by changes of active histone marks (Fig. 7). Furthermore, the significant

enrichment of genetic polymorphisms that have been linked with cardiac disease underlines the pathological relevance of active *cis*-regulatory elements in CMs (Fig. 7).

Several studies highlight the importance of cell-type-specific epigenomic analyses[29]. However, isolation of CMs from human hearts is challenging. In 2009, Bergmann et al. described a technique to isolate CM nuclei using specific protein markers, including PCM1, to study CM proliferation in human hearts[13,14]. The specificity of PCM1 to label postnatal CM nuclei has been characterized extensively for mouse and human hearts[11,12,14]. SIRPA has been used previously as a marker of fetal CMs[15]. Here we use a common marker for both prenatal and postnatal CM nuclei, PLN[30]. The specificity and validity of PLN is strongly supported by immunohistochemical staining[16] and by experimental evidence provided in the present study. Labeling of cardiac nuclei with anti-PLN antibodies enables identification and purification of CM nuclei from prenatal and postnatal human hearts.

Our genome-wide analysis of mCpG identified genic regions with unmethylated CpGs (gUMR)[22,23,33]. In concordance with previous studies[11,31,32], depletion of mCpG affected transcribed regions extending from the TSS toward the 5′ end of genes. Low levels of mCpG correlated with higher levels of active histone marks and gene expression. Analysis of fetal, infantile, and adult non-failing data sets revealed dynamic genic mCpG during prenatal and postnatal development affecting 10% of all genes harboring a gUMR. Alterations within these genes were also present on the layer of histone modifications and gene expression and affect especially genes involved in the maturation of CMs.

Notably, the negative correlation between mCpG and gene expression was not observed for a large group of early developmental genes that were marked with H3K27me3. These genes showed low mCpG despite very low transcriptional activity. Intriguingly, low mCpG was not restricted to genic regions for these genes but extended into surrounding genomic regions, which have previously been termed mCpG valleys[33].

Recent studies show that regions with low mCpG (LMR) have *cis*-regulatory properties[23,24,34,35]. In myocyte nuclei, we

identified >100,000 LMRs. The high degree of cell-type specificity of *cis*-regulatory domains with low mCpG is underlined by the strong enrichment of cardiac transcription factor motifs within these regions. Notably, motifs of different transcription factors were clustered within most *cis*-regulatory regions. The most prevalent combinations include T-box, GATA, or MEF2 motifs. Co-occupancy of these and other cardiac transcription factors has been described previously using a murine atrial cardiac muscle cell line[36] and in vitro differentiated mouse CMs[4]. Alternatively, enhancers can be identified based on chromatin marks. Chromatin mark-based prediction using ChromHMM[37] identified 216 Mb with strong enhancer signature in CMs. *Cis*-regulatory regions like LMRs represent either active enhancers or silent as well as repressive elements. We found a large overlap between *cis*-regulatory regions identified as LMRs and enhancers predicted by chromatin marks.

These findings support that localized low mCpG is a hallmark of *cis*-regulatory regions[23,24,34,38]. Establishment of LMRs has been shown to depend on transcription factor occupancy[24,39]. LMRs show lowest mCpG values at accessible sites as revealed by parallel analysis of mCpG and accessibility in single cells[34,40]. *Cis*-regulatory sites marked by low mCpG can act as enhancers or silencers (Fig. 1f, gray highlighted LMRs). Chromatin state annotation using histone marks is important for the identification of the activation state. Chromatin modifications are deposited at histones flanking transcription factor-binding sites[41]. Combining ChIP-seq and WGBS will likely improve the prediction of localization and activation state of *cis*-regulatory regions. A recent study invented an algorithm integrating ChIP-seq and WGBS data for prediction of tissue-specific enhancers[35].

Comparing our data with a recent study reporting >80,000 putative enhancers (264 Mb) in mouse and human prenatal and postnatal hearts[22] revealed that 17% of enhancers detected in the hearts do not overlap with enhancers detected in CM. These "heart-specific" enhancers were situated near genes associated with non-CM functions. Looking at in vivo confirmed VISTA heart enhancers[22], the overlap with CM enhancers is 90%. Thus our study provides a comprehensive annotation and characterization of *cis*-regulatory regions in human CMs.

During prenatal development and postnatal maturation, >18% of CM *cis*-regulatory regions identified as LMRs showed differential mCpG. Concordant gene expression and gUMR mCpG changes of adjacent genes suggest the functional relevance of these *cis*-regulatory sites. Affected genes were implicated in development and maturation of CMs highlighting the biological relevance of these observations. Regions gaining mCpG showed significant enrichment of motifs related to the GATA, T-box, and TEAD families. A similar combination of motifs has previously been found in murine HL-1 cells[36]. In contrast, loss of mCpG affected regions containing MEF2 and CTF/NF1 motifs. mCpG changes were associated with concordant changes in histone modifications. Notably, regions gaining mCpG during development already showed low levels of enhancer marks at the fetal stage. This may indicate that changes at the chromatin level precede altered mCpG. These differences in kinetics of histone modifications and mCpG have previously been observed during mouse organ development and were termed "epigenetic memory"[42]. In addition, regions losing mCpG were pre-marked with 5hmC, indicating active CpG demethylation[43]. These results clearly show that developmental mCpG of *cis*-regulatory sites is actively modified after lineage decision and orchestrates with dynamic modification of histones and gene expression.

Comparison of mCpG changes during development of fetal and maturation of infantile CMs revealed a continuum with predominant loss of mCpG. In contrast, a recent study[44] analyzing prenatal and postnatal mCpG dynamics in mouse organs,

including heart, reports that continuous loss of mCpG at enhancer regions dominates prenatal development, while gain of mCpG is characteristic for postnatal stages. One reason for this discrepancy could be the different developmental stage of mice and men at birth. Another reason could be a postnatal change in cellular composition. In case of the heart, fetal and infantile hearts predominantly consist of CMs. After birth, CMs loose cell cycle activity. In contrast, endothelial and mesenchymal cell proliferation augments after birth[14]. This results in an approximately 50% lower proportion of CM nuclei in adult as compared to prenatal mouse[11] and human hearts (Fig. 1c). The resulting increasing postnatal cellular complexity may mask cell-type-specific LMRs leading to the predominant detection of a gain of methylation in tissues after birth.

The interplay of epigenetic signatures and gene expression that we observed during development was not evident in chronic heart failure. Only six genes showed differential mCpG at their gene body between non-failing and failing CMs. Of those, none showed a statistically significant differential expression. Also LMRs with *cis*-regulatory properties showed a very low degree of disease-associated mCpG changes. Strikingly, these alterations were not associated with an altered chromatin state and were not found near to differentially expressed genes. Thus mCpG remains relatively stable in chronic heart failure.

However, pathological gene expression in failing CMs was significantly linked with levels of active histone marks. These data are consistent with recent observations on the role of bromodomain proteins in the heart, which recognize acetylated histones[45,46]. Blockade of the bromodomain protein Brd4 by small molecule compounds inhibited pathological gene expression and progression of heart disease in a mouse model of cardiac pressure overload and in CMs in vitro[45,46]. A recent series of publications[47–49] reports a regulatory role of the histone mark H3K9me2, a mark not studied in this project. Future studies are necessary to show whether the proposed mechanisms are involved in human heart failure.

Genome-wide association studies for multiple diseases have identified a plethora of disease-associated variants in the non-coding genome[50,51]. Many of these genetic variants were mapped to loci with regulatory properties, including enhancers[52]. Recent comprehensive studies have generated genome-wide enhancer maps of heart tissue and have linked specific enhancer regions with genetic variants associated with myocardial repolarization and conduction[28] and other forms of cardiac disease[22]. However, these studies do not address in which cardiac cell type and developmental or disease state disease-associated enhancers exist. The present study identified >100,000 LMRs with *cis*-regulatory signatures. Genetic variants that have previously been linked with cardiac arrhythmia, coronary, or congenital heart disease were enriched in these regions in CMs. Notably, SNP-bearing LMRs were strongly associated with genes, which have previously been identified in genetic forms of cardiac channelopathies, including pacemaker channels (*HCN4*—Brugada syndrome), multiple K$^+$ channels (*KCNE1*, *KCNH2*, *KCNJ2*, *KCNQ1*—long and short QT syndromes), and voltage-gated Na$^+$ channels (SCN5A—Brugada syndrome, sick sinus syndrome)[53]. Further studies are needed to directly link enhancers containing genetic variants with CM genes.

The present study identifies mCpG and histone modifications as connected or separate layers of epigenetic regulation during human CM development and disease, respectively. This detailed insight into the CM epigenome in intact human hearts will be important for several areas of research. First, this CM epigenome may serve as a roadmap for further studies in embryonic stem cells or induced pluripotent stem cells to generate mature CMs in vitro for cell therapy of heart disease and for direct

reprogramming of cells into CM in vivo. Second, our epigenetic data enable functional annotation of non-coding regions of the genome in CMs. This will be important to unravel the genetics of cardiac disease in non-coding regions of the genome. Further epigenetic studies of other cardiac cell types will help to better understand the contribution of individual cell types to cardiac disease. Third, this CM epigenome provides comprehensive insight into the molecular marks that are associated with physiological and pathological gene expression in CMs. Future studies mapping the three-dimensional architecture, protein complexes, and non-coding RNAs will help to develop new strategies for treatment of heart disease.

## Methods

**Human cardiac biopsies.** LV biopsies from male hearts were used for CM nuclei isolation. These investigations were approved by the ethics committees of the Mount Sinai School of Medicine (New York, USA) and the Universities of Freiburg, Jena and Munich (Germany) (Suppl. Table 1) with informed consent of human participants. All samples retrieved during interventions (fetal and failing as well as rejected donor hearts) were immediately flash frozen and stored at −40 to −80 °C. Tissue from accidentally killed patients was flash frozen during the autopsy not later than 24 h after death. To comply with the ethics committee approval, we unassigned patient information and genomic sequence information (Supplementary Data 1 and 2).

**FACS of CM nuclei.** All steps were performed at 4 °C to ensure integrity of chromatin and RNA. For nuclear RNA isolation RNAsin (80 U/ml, Promega) was added to all buffers. Isolated nuclei were stained in 500 μl staining buffer (phosphate-buffered saline (PBS) containing 1 % bovine serum albumin (BSA), 22.5 mg/ml glycine, 0.1% Tween 20) using anti-PCM1 (1:500, HPA023370, Sigma) and anti-PLN antibodies (1:500, A010–14, Badrilla) for 30 min. For isotype control stainings, we used primary antibodies lacking target specificity (1:1000, anti-mouse, 554121, BD; 1:1000, anti-rabbit, Z25308, Life technologies). Subsequently, the corresponding Alexa488- and Alexa568-labeled secondary antibodies (1:1000, A11029 and A11011, Invitrogen) were added. After 30 min of incubation, nuclei were pelleted by centrifugation (1000 × g, 5 min) and resuspended in 1 ml PBS containing 1 mM ethylenediaminetetraacetic acid (EDTA). Nuclei were filtered (CellTrics 30 μm, Sysmex) and incubated with Draq7 (final concentration 2.25 nM, Cell Signaling) for 10 min. Nuclei were analyzed (Bio-Rad S3, Bio-Rad; LSRFortessa, BD) and sorted by flow cytometry (Bio-Rad S3, Bio-Rad).

**FACS of fetal CMs.** Human fetal heart tissue was cut into small pieces and incubated in collagenase type II (1 mg/ml; Worthington) in Hanks solution (NaCl, 136 mM; NaHCO$_3$, 4.16 mM; NaPO$_4$, 0.34 mM; KCl, 5.36 mM; KH$_2$PO$_4$, 0.44 mM; dextrose, 5.55 mM; HEPES, 5 mM) for 6 h at 37 °C with gentle shaking. After incubation, cells were centrifuged (250 × g, 5 min) and the supernatant was removed. BSA 1 mg/ml in PBS-Ca$^{2+}$/Mg$^2$ solution was added to the cellular pellet and was pipetted gently to dissociate the cells. After dissociation, cells were filtered and used for analysis. Cells were stained in ice for 1 h at a concentration of 2.5 × 10$^6$ cells/ml with anti-SIRPA-IgG-phycoerythrin-Cy7 (clone SE5A5; BioLegend; 1:100) and anti-CD90-allophycocyanin (clone: 5E10; BD Pharmingen; 1:100). For FACS, the cells were sorted at a concentration of 10$^6$ cells/ml in PBS-Ca$^{2+}$/Mg$^2$ with 10% fetal bovine serum using an Aria II cell sorter (BD Biosciences). To prevent cell death due to pressure and sheer stress, all sorts were performed with a 100-μm nozzle.

**Chromatin immunoprecipitation sequencing.** Isolated CM nuclei were fixed with paraformaldehyde (final concentration 1 %, Thermo Fisher) for 2 min at room temperature. Fixation was quenched by adding glycine to a final concentration of 0.125 M. Nuclei were washed twice with washing buffer (50 mM Tris-HCl pH 8.0, 10 mM EDTA) supplemented with protease inhibitors (cOmplete Protease Inhibitor Cocktail, Roche). After addition of sodium dodecyl sulfate to a final concentration of 1%, nuclei were incubated for 10 min. Chromatin was sonicated using a Bioruptor (Diagenode) with 30 cycles (30 s on followed by 30 s off; high energy). After centrifugation (13000 × g, 10 min), the chromatin containing supernatant was used for ChIP experiments. The amount of chromatin (measured as DNA) was quantified using a Qubit 2.0 fluorometer (Life Technologies) and the corresponding Qubit dsDNA HS Assay Kit (Life Technologies). In all, 100–200 ng of isolated chromatin was used for ChIP experiments. Immunoprecipitation, reversal of cross-linking, and DNA purification were performed using the ChIP-IT High Sensitivity Kit (Active Motif) following the manufacturer's manual and the antibodies listed in Supplementary Table 1. The eluted DNA (200 μl) was concentrated by evaporation at room temperature using a vacuum concentrator (Eppendorf) to 55.5 μl. Libraries were prepared using the NEBNext Ultra DNA Library Prep Kit for Illumina (NEB) according to the manufacturer's manual omitting the size selection. Library amplification was monitored after addition of Eva Green (1:20, #31000, Biotium)

by quantitative PCR (qPCR) using a real-time PCR cycler (Stratagene) and stopped after the turning point of the amplification curve was reached. Libraries were sequenced on Illumina sequencers in paired-end mode.

**Nuclear RNA sequencing.** Nuclear RNA was extracted from sorted nuclei using either the RNeasy Micro Kit or the AllPrep DNA/RNA Micro Kit (Qiagen) including on-column DNAse digestion. RNA was reverse transcribed and amplified with the Nugen Ovation RNA-seq System V2 (Nugen). The resulting cDNA was fragmented (Bioruptor, Diagenode). Sequencing libraries were constructed using 100 ng fragmented cDNA using the NEBNext Ultra DNA Library Prep Kit (NEB). The necessary PCR cycles were determined by qPCR after addition of Eva Green (Biotium). RNA-seq libraries for direct comparison of PLN- and PCM1-positive nuclei were prepared using the NEBNext Ultra RNA Library Prep Kit for Illumina (NEB) in combination with the RiboMinus Eukaryote System v2 (Life Technologies)[12]. All libraries were sequenced on Illumina sequencers in paired-end mode.

**Whole-genome bisulfite sequencing.** WGBS-seq libraries were constructed from 250 ng DNA using the Ovation Ultralow Methyl-Seq Library System (Nugen) or from 0.5–1.5 μg DNA using the NEXTflex Bisulfite-Seq Kit (Bioo). For small DNA amounts between 5 and 30 ng from sorted fetal CMs, a tagmentation-based protocol (T-WGBS)[54] was chosen. Libraries were sequenced on Illumina sequencers in paired- or single-end mode.

**5hmC sequencing.** DNA was extracted from sorted nuclei using the AllPrep DNA/RNA Kit (Qiagen). DNA was sheared in a volume of 100 μl at low intensity for 30 cycles (30 s on followed by 90 s off) using a Bioruptor (Diagenode). Sheared DNA was processed using the hydroxymethyl Collector–Seq Kit (Active Motif) according to the manufacturer's instructions. The resulting DNA was used for library generation using the NEBNext Ultra DNA Library Prep Kit (NEB). Libraries were sequenced on Illumina sequencers in paired-end mode.

**Gene ontology (GO).** To identify enriched GO terms in the categories "biological process", "cellular component" and "molecular function" ClueGO[55] was used. Bonferroni step-down correction was performed for multiple testing-controlled p-values. The GO term connectivity threshold was adjusted to 0.3 (kappa value) and only pathways with a p-value ≤ 0.01 were considered as significant. The resulting terms were functionally grouped and illustrated, whereby the most significant term in each group was selected as leading term.

**Processing of sequencing reads and mapping.** All bioinformatics tools used to analyze sequencing data were integrated in the Galaxy platform[56]. Quality and adapter trimming of sequencing reads was performed prior to mapping to remove low quality reads and adapter contaminations. RNA-seq data were mapped to the human genome (hg19) using Tophat2[57]. ChIP-seq and 5hmC-seq data were mapped using Bowtie2[58] and WGBS-seq using Bismark[59]. PCR duplicates were removed using SAMtools[60]. Quality and adapter trimming of sequencing reads was performed prior to mapping to remove low quality reads and adapter contaminations. RNA-seq data were mapped to the human genome (hg19) using Tophat2[57]. PCR duplicates were removed using SAMtools[60].

**RNA-seq data analysis.** Owing to the high number of intronic reads arising from unspliced RNAs in nuclear RNA-seq data[12,61], we used intronic and exonic regions of coding genes for gene expression analysis. Mapped RNA fragments were further processed using Cuffnorm[62] to calculate FPKM values as an estimate of transcript expression. For differential gene expression, HTSeq count[63] and Deseq2[64] were used. A q-value of <0.05 was considered as significant. Genes with expression values of <3 FPKM in all the compared groups were excluded from the differential gene expression analysis. We used Cufflinks[62] to determine the main transcription start (TSS) and stop site (TES) of each coding gene using merged RNA-seq data from all assessed stages. To exclude biases, we analyzed alternative usage of TSS and TES sites using Cuffdiff[62] and did not observe significant differences between fetal, infant, adult non-failing, and adult failing CMs.

**Correlation of gene expression and epigenetic marks.** For a comparison of differential gene expression with results obtained from ChIP-seq and WGBS, genes differentially regulated in failing as compared to non-failing hearts were ranked according to fold changes (log2(F/NF)). This ranking was applied to data obtained from ChIP-seq and WGBS. WGBS data were used to calculate differential mCpG of gUMRs and genic regions of the ranked genes (F–NF). The ratio of ChIP-seq data (F/NF) was calculated using bamCompare[65]. For quantification of H3K27ac, H3K9ac and H3K4me3 promoters (TSS ± 1 kb) and for H3K36me3, H3K9me3, H3K27me3, and H3K4me1 genic regions (TSS to TES) were used.

**ChIP-seq and 5hmC data analysis.** Sequencing data were processed using DeepTools[65]. For comparison of different data sets, read counts were normalized to the geometric mean of the genome-wide read count. This normalization was performed for the different analyzed features (i.e., gUMRs, LMRs). To combine

results of different histone modifications, the normalized ratios of the individual marks were averaged on a logarithmic scale. For visualization, we merged the results of biological replicates.

**ChrommHMM**. ChromHMM[37] was used to learn a 10-state model predicting chromatin states based on ChIP-seq data from non-failing adult CMs. For the annotation of chromatin states in CMs, we used ChIP-seq data generated for H3K27ac, H3K9ac, H3K36me3, H3K4me1, H3K4me3, H3K9me3, and H3K27me3. For comparison of chromatin states in adult CMs and heart tissues, we used the same ChIP-seq data from adult non-failing CMs except for H3K9ac, since these data are not available from Encode or Roadmap[18–20]. The derived states were classified as enhancer (H3K4me1, H3K9/27ac), promoter (H3K4me3), transcribed chromatin (H3K36me3), bivalent chromatin (H3K4me3, H3K27me3), hetero-chromatin (H3K9me3), polycomb repressed chromatin (H3K27me3), and silent undefined chromatin (no enriched chromatin mark).

**External data**. Cardiac enhancers (predicted VISTA heart enhancers) derived from >35 human and mouse H3K27ac and p300 data sets were obtained from Dickel et al.[22]. In vivo confirmed VISTA heart enhancers were also obtained from Dickel et al.[22]. For comparison of CM data with published heart data sets, we reanalyzed WGBS (adult 34 years, Encode number ENCSR579AXB; 3 years' infant LV, ENCSR012TGL; 101 day fetal heart, ENCSR699ETV), and ChIP-seq data (34 years' adult LV: H3K9me3, ENCSR176KNR; H3K4me1, ENCSR111WGZ; H3K36me3, ENCSR434MDA; H3K27ac, ENCSR150QXE; H3K4me3, ENCSR487BEW, H3K27me3, ENCSR503YOF) generated by the Encode and Roadmap consortia[18–20].

**mCpG analysis**. mCpG data were extracted from mapped reads and data of both strands were merged using MethylDackel (https://github.com/dpryan79/MethylDackel). For segmentation[20,21], we merged the replicates of the different stages to increase the coverage. In brief, genomic regions with highly disordered mCpG (PMRs) were identified using a 101 CpG sliding window with a step size of 1 CpG. A two-state Hidden-Markov model identified PMRs based on the deviation of methylation level distributions from the typical polarized distribution, which favors high and low methylation. Neighboring PMR segments were fused. After masking of PMRs, we smoothed the mCpG data using a running window of three CpGs to reduce noise. To identify suitable parameters (minimal number of CpGs and mCpG cut-off) for the identification of hypomethylated regions, we calculated the false discovery rate (FDR) by comparing the number of identified regions in the original and a randomized methylome. For all methylomes generated in this study, at least 3 CpGs and a mCpG of ≤50% was chosen as parameter for the detection of hypomethylated sites. These parameters ensured in all assessed methylomes an FDR < 5%. We used a cut-off of 30 CpGs to discriminate LMR and UMR (LMRs < 30 CpGs, UMRs ≥ 30 CpGs)[20,21]. This cut-off clearly separated CpG-low regions with methylation levels between 10 and 50% (LMRs) from un-methylated CpG-rich regions (UMRs)[20,21].

Since the segmentation result is influenced by the local sequencing depth and surrounding mCpG variability, we merged the segments of the different stages using MergeBED[66] to obtain a comprehensive set of CM LMRs, UMRs, and PMRs. These regions were used for all further steps. The high degree of UMRs overlapping genic regions prompted us to define gUMRs. Since genic demethylation is initiated at the TSS and progresses toward the 5′ region of the gene, gUMRs overlap with the TSS. Only genes spanning at least 1.5 kb outside of CpG islands were used for this assignment. Finally, CpGs overlapping with annotated CpG islands[67] were removed to obtain gUMRs. Genes with distinct low genic mCpG fulfilled the following criteria. They harbor a gUMR of at least 5 kb and/or the gUMRs overlap with at least 25 % of the gene. An additional prerequisite is that the gUMRs consist of at least 10 CpGs. Fully methylated regions (FMRs) represent genomic regions not overlapping with CpG islands, LMRs, UMRs, and PMRs.

Replicate-based differential methylation analysis (replicate DMR) was performed using Metilene[68] for FMRs, CpG islands, LMRs, gUMRs, and PMRs. Since genic regions downstream of gUMRs were frequently partially methylated too, PMRs overlapping a single gene to at least 75% were removed from differential methylation analysis. For the analysis of differential mCpG, we used the results obtained from the individual biological replicates. A difference of >10% and a p-value ≤ 0.001 was considered as significant. A minimal coverage of one in all compared samples was required to include a CpG in the statistical analysis. Biological replicates with a median CpG coverage of ≤4 were excluded from the statistical analysis.

For the identification of DMRs between adult heart tissue and CMs, we used the binary segmentation algorithm implemented in Metilene[68]. For this analysis, we used Encode data from human left ventricle and compared it with the merged data obtained from CM nuclei from non-failing adult hearts. A difference of >10 % and a p-value ≤ 0.001 was considered as significant. DMRs spanning <10 CpGs were rejected.

For visualization, we merged the results of biological replicates and smoothed the data using a running average of three CpGs.

**General bioinformatic analysis**. We used Homer[69] to perform a de novo motif analysis using the "size given" option. GREAT[27] was used to identify genes in proximity (≤10 kb) to LMRs and to annotate genes within regulatory domains of SNPs. DeepTools[65] and HiCExplorer[70] were used to display traces and heatmaps and to perform cluster analysis. Annotations and genome files (hg19) were obtained from the UCSC[71]. Prism 5 (GraphPad) was used for statistical analysis and graphs.

**SNP identification**. Disease-associated SNPs were obtained from the NHGRI-EBI GWAS catalog for different traits on September 30, 2016[72]. Search terms were: cancer, digestive system disease, immune system disease, metabolic system disease, nervous system disease, and hypertension. In addition, we searched and combined loci retrieved for the search terms congenital heart and heart malformation (combined term: congenital heart disease), myocardial infarction, and coronary heart disease (combined term: coronary heart disease) as well as heart rate, PR interval, QT interval, and cardiac arrhythmia (combined term: cardiac arrhythmia). Duplicate SNPs were removed. SNAP[73] was used to retrieve proxy SNPs in linkage disequilibrium ($R^2 > 0.8$, CEU population, 1000 Genomes, Pilot 1)[72] with the lead SNPs.

**Data availability**. All sequencing data sets reported in this manuscript are deposited in the Short Read Archive at the National Center for Biotechnology Information under the BioProject ID PRJNA353755. Additional data that support the findings of this study are available from the corresponding author upon request.

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

## Acknowledgements

We thank Claudia Domisch for excellent technical support. We thank the European Molecular Biology Laboratory GeneCore, Genomics Core Facility staff (Heidelberg, Germany) for performing MethylC-seq, and the Deep Sequencing Facility, Max Planck Institute of Immunobiology and Epigenetics (Freiburg, Germany) for sequencing. We thank Andreas Würch for FACS analysis support. This study was supported by the Deutsche Forschungsgemeinschaft SFB 992 project B03 (to L.H.), PR 1668/1–1 (to S.P.) and GI 747/2−1 (to R.G.) as well as the BIOSS Centre for Biological Signalling Studies (to L.H.).

## Author contributions

Conceptualization: L.H., R.G.; methodology: R.G., M.S., B.D.G., S.M.-N., S.P.; software: B. A.G.; investigation: R.G., M.S., S.P., D.K., T.G.N., P.S., S.M.-N., D.W., C.B., M.D., H.L.; analysis: R.G., M.S., L.H.; resources, A.R.J., C.B., B.D.G., T.D., M.K.; visualization: R.G., M.S., L.H.; writing—original draft: R.G., L.H.; writing—review & editing: M.S., R.G., L. H., B.D.G., M.K., R.B., T.G.N.; supervision: L.H.

## Additional information

**Competing interests:** The authors declare no competing financial interests.

**Reprints and permission** information is available on line at http://npg.nature.com/reprintsandpermissions/

Ralf Gilsbach[1], Martin Schwaderer[1], Sebastian Preissl[1], Björn A. Grüning[2], David Kranzhöfer[1], Pedro Schneider[1], Thomas G. Nührenberg[1,3], Sonia Mulero-Navarro[4], Dieter Weichenhan[5], Christian Braun[6], Martina Dreßen[7,8], Adam R. Jacobs[9], Harald Lahm[7,8], Torsten Doenst[10], Rolf Backofen[2], Markus Krane[7,8,11], Bruce D. Gelb[4,12] & Lutz Hein[1,13]

[1]Institute of Experimental and Clinical Pharmacology and Toxicology, Faculty of Medicine, University of Freiburg, 79104 Freiburg, Germany. [2]Bioinformatics Group, Department of Computer Science, University of Freiburg, 79110 Freiburg, Germany. [3]Department for Cardiology und Angiology II, University Heart Center Freiburg ● Bad Krozingen, 79189 Bad Krozingen, Germany. [4]The Mindich Child Health and Development Institute, Icahn School of Medicine at Mount Sinai, New York, NY 10029-6542, USA. [5]Epigenomics and Cancer Risk Factors, German Cancer Research Center (DKFZ), 69120 Heidelberg, Germany. [6]Forensic Institute, Ludwig-Maximilians-University, 80046 Munich, Germany. [7]Department of Cardiovascular Surgery, German Heart Center, Technische Universität München, 80636 Munich, Germany. [8]Insure (Institute for Translational Cardiac Surgery), Department of Cardiovascular Surgery, German Heart Center, Technische Universität München, 80636 Munich, Germany. [9]Department of Obstetrics, Gynecology and Reproductive Science, Icahn School of Medicine at Mount Sinai, New York, NY 10029, USA. [10]Department of Cardiothoracic Surgery, Jena University Hospital, Friedrich-Schiller-University, 07740 Jena, Germany. [11]DZHK (German Center for Cardiovascular Research) - Partner Site Munich Heart Alliance, Munich 60046, Germany. [12]Department of Genetics and Genomic Sciences & Department of Pediatrics, Icahn School of Medicine at Mount Sinai, New York, NY 10029-6574, USA. [13]BIOSS Centre for Biological Signalling Studies, University of Freiburg, 79104 Freiburg, Germany

