## [Peer Review File · Nature Communications]

Reviewers' comments:

Reviewer #1 (Remarks to the Author):

The authors describe the human cardiac myocyte epigenome at three stages during life (fetal, infant and adulthood) and at one diseased condition of end stage heart in adults. The study is of high interest to the cardiac community however, the manuscript lacks scientific clarity, reads very polished, holds wrong estimations and statements.

A. Major general points:

1) The author studied 6 fetal, 4 infants, 5 non-failing and 6 failing human cardiac samples. The undertaken analysis of WGBS, RNA-seq, histone-ChIP-seq and 5hmC were performed at different samples of different cases. This fact is only given in the Supplemental Table S1, not as a part of the general manuscript. The manuscript is written such that one assumes that all experiments were performed on all samples.

Here is an example given for the fetal samples, which holds true also for all other stages: two cases had WGBS, two different cases had histone-ChIP-seq (H3K9me3 only one sample), one different case had 5hmC; and 4 cases not overlapping with the WGBS cases had RNA-seq.

In the introduction, the authors write that data produced in the past by considering total biopsies and not selected cardiomyocytes are misleading and simply speaking bad data. I agree that isolated cardiomyocytes are better, but, data generated at different levels (RNA-seq, ChIP-seq, WGBS) as in previous total samples studied by others where frequently generated from one sample, and moreover these studies considered a statistically significant number of samples.

The study presented in the manuscript has a clear inter-individual bias, as data that are connected and analyzed in relationship to each other (for example RNA-seq related to WGBS) are generated from samples of different individuals and moreover, the sample number of two does not allow any statistical evaluation.

The manuscript should include an overview explaining which experiments were performed from which case in addition to the supplemental table S1.

Case descriptions for all stages given in table S1 would clearly benefit from more details of the phenotype, in particular as the analysis was performed across samples and their comparability is essential. For example, in four fetal cases the age is given in the range of 18-21 weeks, moreover as abort reason it is just stated "non-cardiac". A clinical description would be helpful as in case of insufficient hemodynamic supply, the heart could be affected even if the cardiac structure is grossly normal. In case of failing hearts, where these dilated cardiomyopathies, hypertrophic cardiomyopathies or any else?

In summary, the cross-sample analysis of related data in combination with small sample numbers of each experiment provides a statistical problem. Here the comparison of selected data with previously obtained data in non-cardiomyocyte isolated samples might be helpful to bring confidence. Publications suitable for this are cited in the manuscript and moreover a WGBS tracks of a human young as well as an adult cardiac sample was generated by the Epigenome Consortium.

2) The description and presentation of the computational analysis is grossly insufficient. There are no quality plots or quality data provided in the supplement. Not even a simple PCA plot showing the relationship of the different samples to each other is provided. The presented inter-individual analysis as described above is essentially depending on a clear co-clustering of related samples. The description of the computational analysis is as bad, as not even the methylation levels used to define the used categories four UMR, LMR, PMR and FMR can be found. At least I searched hard

and did not find them. It is insufficient to reference two articles from 2011 and 2013 for this. Moreover, different methylation categorizations are currently in use, e.g. the Meissner group (Tsankov et al. Nature 2015) uses just three categories, namely, unmethylated regions (UMR, where $0 \leq \text{UMR} \leq 10\%$ methylation), intermediate methylated region (IMR, where $10\% \leq \text{IMR} \leq 60\%$ methylation), highly methylated region (HMR, where $60\% \leq \text{HMR} \leq 100\%$).

A rationale for the applied categories would be helpful. Moreover, why not using DMRs? For time-series data, as in the manuscript, the application of DMRs is a common approach (Schultz et al. Nature 2015; Ziller et al. Nature 2013).

DMRs provide a better quantification of methylation changes between different samples than used qualitative categories, and therefore would be a better option for relating alterations of DNA methylation and Histone marks (quantitative ChIP-seq data) to each other, which is an essential part of the study.

3) The study does not take any advantage of having three time points throughout life, but this sample collection could be the strength of the study. Why are methylation information in Fig 7 summarized for all samples? What are the differences between neighbored stages? How does DNA methylation change over time.

4) The author misuses the term "development". This term should be applied for the comparison between fetal and infant samples. The term "aging" should be used for the comparison between infant and adult stages. This needs to be corrected throughout the manuscript.

5) Moreover, the authors mainly compare fetal to adult samples, without consideration of the interval of infant samples. This is scientifically not sound, a comparison between 18-21 weeks old fetal cardiac samples to 50-60y old adult samples has little information. The cardiovascular fetal circulation is totally different from an infant circulation and a comparison between these would be very interesting as it addresses this hemodynamic adaptation. I skip any explanations here of the circulatory differences as this is common knowledge. Comparing infant to aged adult samples is also very informative for the study of cardiac aging.

However, the majority of results provided compare fetal to adult cases and therefore mix hemodynamic and aging processes, this is of very little scientific value. Thus, a large part of the result section should need to be reanalyzed to correct this and the manuscript to a large extent needs to be re-written. The project summary might even be very different.

B. Selected major specific point:

1) Figure 1c, shows the proportion of cardiac myocyte nuclei in fetal (Fe, n=3), infantile (I, n=5), adult non-failing (NF, n=5) and adult failing (F, n=5) LV tissue. In the figure are five data points for each of the samples type shown. How can this be, if Fe has only a n=3 as stated in the figure legend, moreover, in table S1 are only four infantile cases/samples described, how can five infantile samples be part of this figure and its legend?

2) In the manuscript, one central claim, presented twice is that: "Remarkably, 33% of the cardiac myocyte CpG methylome was shaped after birth between infant and adult stage."

This sentence is totally incorrect at least considering the data presented in the manuscript. Correct is that 33% of the dynamic LMRs differ between infant and adult stages; dynamic LMRs represent only 15% of all LMRs. Moreover, the cardiac methylome consists of UMRs, LMRs, PMRs and FMRs. Therefore, correct is that below 0.1% of the cardiac myocyte CpG methylome is shaped after birth between infant and adult stages, and this is not surprisingly.

C. Selected minor points:

1) Figures could be homogenized, for example in Fig 2c, genes are given for gUMR whereas in the comparable analysis of LMRs in Fig 4a, LMRs regions are given.

2) Statistical values are frequently missing, it is just stated that something would be significant, see for example Fig 4a.

Reviewer #2 (Remarks to the Author):

This manuscript by Gilsbach et al. maps the epigenome (7 histone marks), methylome (whole-genome bisulfite sequencing) and transcriptome (RNA-Seq) of human cardiac myocytes during development of the heart from fetal to adult. They refined methods for purifying cardiomyocyte nuclei from tissue by flow cytometry and analyzed cardiac myocytes at 3 time-points: fetal, infantile and adult along with comparing between adult non-failing and terminally failing hearts. They first segmented the genome according to its methylation status and then by comparing the nearest-gene annotation and epigenetic signature determined that low-methylated regions are putative enhancers whereas un-methylated regions are present in gene bodies and CpG islands. They found that methylation status and histone marks at gene bodies are correlated with gene expression for the subset of genes that are developmentally regulated during myocyte maturation. In their study of pathologically important genes for heart failure however they found that while active histone marks could to some degree explain the change in gene expression, the same was not true for inactive marks and CpG methylation. They then repeated this analysis for the low-methylated regions (putative enhancers) finding similar interplay between changing methylation, histone signature and gene expression during development. They again could not find any pattern for disease-associated genes. They also found that cardiac disease associated genetic polymorphisms were enriched at these low-methylated regions.

The authors have carried out a substantial body of work and generated datasets that will be valuable to those studying heart development, heart failure, or cardiac gene regulation. The methods for isolation of cardiomyocyte nuclei will also be valuable. However, the major conclusion of the paper that methylation status and the epigenome are closely linked to changes in gene expression during CM maturation, is not novel and has been shown in other developmental systems. They also studied changes during heart failure but did not have major novel findings.

Major concerns:

1. Novelty, as noted above.
2. The authors reported a similar study of murine cardiomyocytes in Nat. Comms 2014. There is little comparison of the results of this study of human cardiomyocytes to the prior work on murine cardiomyocytes. Were the major findings in each system consistent? What is the conservation of epigenetic marks, especially DNA methylation? Aside from providing information on human cardiomyocytes, what advances did this study make over the 2014 study?
3. The authors report having done replicates, but do not provide information about the reproducibility of the data. How were replicates combined?
4. The authors should provide more data about the human myocardial samples, especially the failing heart samples. There is considerable heterogeneity in human heart failure, and many causes of heart failure. The cardiac diagnoses should be provided. The authors should consider the possibility that negative findings in heart failure were due to heterogeneity and lack of sufficient statistical power. Was there more intersample heterogeneity in the failing samples than the other groups? How would this impact the analysis and the ability to detect differences? Along these lines, on page 8 the authors show that gene expression in disease is not correlated with DNA methylation or histone repressive marks and therefore conclude that these marks are stable. However, lack of correlation cannot be equated to stability.
5. The authors started by segmenting the data according to methylation status. They did most of their analysis on gene-unmethylated regions (which are basically gene bodies) and low-methylated regions (putative enhancers). The advantage of this approach over chromatin-based segmentation to divide the genome into promoters, enhancers, inactive and other regulatory regions is not clear. Maybe the authors can show some type of clustering analysis to illustrate why this is a superior approach. Does adding DNA methylation to histone data in ChromHMM or similar modeling

substantially influence segmentation into functional elements?

6. The authors should compare the putative enhancer regions identified in their study to those identified in other studies of human heart enhancers (e.g. May et al., Nat Genet; Dicket et al, Nat Comms). What is the sensitivity and specificity of detecting regions with transgene-proven heart enhancer activity (e.g., VistaEnhancer database)? Does the methylation data improve detection of enhancers?

7. The authors propose that low methylated regions (LMRs) are cis-regulatory elements. However, only half of the LMRs are marked by H3K27ac. What about the other half? Are H3K27ac+ LMRs associated with greater gene expression than H3K27ac- LMRs? What fraction of H3K27ac regions have LMRs?

8. For transcriptome analysis the authors use cufflinks suite. Though cufflinks is widely used, it is known to underestimate differentially expressed transcripts. Maybe the authors can redo transcriptomic analysis using a second method (e.g RSEM, EBSeq if the authors are interested in transcript level analysis or DESeq2 if gene-level) and show the agreement between methods to ensure that any differential expression analysis is not driven by the choice of methods. Also in the methods section on page 20, line 7: the authors talk about selecting the main cardiac myocyte isoform of each gene. It is not clear how this was determined. If this was based on expression it can be problematic as isoforms might be differentially expressed during development. Due to these concerns it seems best to carry out gene-level differential expression analysis with a count-based method.

Minor concerns:

1. On page 6, line 19; its not clear whether the genes are expressed in prenatal, postnatal cardiomyocytes or either.
2. On page 6, line 19; the authors should mention some quantitative statistic detailing the level of correlation between CpG methylation and gene expression.
3. On page 7 line 1; the authors talk about 2 groups of genes associated with different signature and enriched for different GO terms. The list of genes in both group I and group II can be informative for the field and should be detailed in a supplementary table along with at least the top 10 Go terms enriched
4. On page 8, line 17; The exact methodology for ranking the genes should be detailed in the methods section.
5. On page 9, line 2; The list of LMRs with decreasing or increasing CpG methylation during development could be informative for the field and should be detailed in a supplementary table.
6. On page 19, line 27; Usually for ChIP-Seq, it is preferred to keep only those reads that map uniquely to the reference genome. In Bowtie this can be done with -m 1 option. However Bowtie2 doesn't have this feature and requires additional processing steps downstream. Did the authors do additional processing?
7. On page 19, lines 30-32 are repeated

Reviewer #3 (Remarks to the Author):

Gilsbach et al present a paper describing the epigenetic status and transcription factor networks at various stages of human cardiac pre-natal and post-natal development and in chronic heart failure.

The aim of this study is of great interest and novelty in the field and it has the advantage of studying purified cardiomyocytes (nuclei) obviating to the confounding data obtained by whole tissue studies.

The authors' conclude that during differentiation of adult cardiomyocytes, the induction of specific genes (i.e. MYH6, RYR2; TNNI3) or repression of others (MYL4, NPPA, TNNI1) depend on the combination of methylation levels at enhancers and gene bodies as well as histone marks while . In contrast, induction of fetal genes in cardiomyocytes of failing hearts is associated to active histone marks but no or little DNA methylation changes. Importantly, the authors have identified

SNPs, previously associated to cardiac disease, in enhancers displaying low methylation. Overall, the paper is well written and the presentation of the data is logical although the reviewer has some suggestions and comments.

The authors used samples from four groups of donors with 4-6 donors/group. It would be important to summarize in brief the characteristics of the donors shown in table S1 in the results section. The cohort used for the present study is good, however, it would be important to have a better understating of how the donors were selected and their clinical history including for example history of congenital diseases, cancer, infections, diabetes and any previous or current treatment. The reviewer would like to know the time frames between deaths, tissue harvest and tissue processing from all the samples of each group. This information should be taken in account as a potential technical factor influencing the results.

It would be useful, for clarity, to spell out in the results (not just the discussion) that the nuclear marker pericentriolar material 1 (PCM-1) enriches for cardiomyocytes' nuclei not any nuclei (Bergmann et al 2010, 2011 and 2015). Indeed, this would help interpreting Figure S1 and S2. Figure S1 is hard to interpret because of the lack of gates or quadrants and the use of a different scale for the axes. Assuming that isotype controls were used for each sample, the reviewer suggests to add the isotype controls to Fig S2, using same scale axis for each set in conjunction with gates or quadrants. Additionally, the reviewer would like to see a more comprehensive representation of these data: in Fig 1e only the percentage of cardiomyocytes nuclei is shown, it would be useful to see in a graph the summary of the percentages of the other populations in the various groups of patients, not just a representative dot plot. This would be important in view of the abundant PCM1+PLN- cells seen in infants samples (Fig S2) as this might represent a specific set of cardiomyocytes or a specific stage of differentiation. Interestingly, no PCM1-PLN+ population appears in the dot plots shown in Fig. S2. Additionally, data from Bergmann et al 2015 suggests that the number of endothelial and mesenchymal cells in the myocardium increases in adult hearts, thus, the reviewer would like to see whether there is a consistent increase in the PCM1-PLN- population from fetal to adulthood and if these represent endothelial/mesenchymal cardiac cells and/or nuclei.

The reviewer is intrigued by the comments on page6 lines 8-14 regarding demethylation, however, the data presented in Fig S8 could be presented in a more convincingly way to support the statements. In essence, where and how do the authors prove that UMRs mirror demethylation? Are the authors presuming this based on the induction of a given gene expression or do they have direct evidence? In addition, how do the authors address and explain the potential differences seen at the various stages analysed from a statistical and biological point of view (see for example Fig S8 j, k and i)?

The scheme in Fig 7 is very useful in summarizing the message, the legend could benefit from some clarification as it is unclear whether UMR and LMR that in the brackets refer to the induced genes or repressed genes. Also, the scheme should highlight better the differences between gene body and enhancers methylation. In addition, would it be appropriate to talk about gene induction albeit demethylation is deficient in the failing hearts? Are the levels of genes expressed in these circumstances lower compared to a situation where you have UMRs plus active histone marks? How do you expect the expression to be in the fetal and earlier phases of cardiomyocytes differentiation and function in the same patients?

It would be intriguing to generate iPS-cells from failing patients and test the epigenetic and transcriptome in relation to these SNPs during various stages of cardiomyocytes differentiation. The authors show original traces (i.e. Fig1f and 3a): can they clarify how these data are displayed? Do they always show the same sample/donor for each group and could they clarify which sample is it? Could they also clarify in the instances where they pool data from all the donors of each group as described in TableS1?

Minor issues

Page 3, line 26: although this is stated elsewhere, the reviewer suggest to add a statement describing what is changing in the pathological conditions studied (active histone marks) given that no CpG methylation modifications were detected.

Fig1 e and especially f: the colour used for 'Fe' samples is hardly distinguishable from 'I' and 'NF' Fig S8, it is difficult to distinguish the colours, this could be obviated by increasing the thickness of

the lines and/or changing the colours.

The font of main and supplemental figures' labelling is in most cases too small and hard to read.

In the figures, in several instances the lines of the curves are too thin.

Response to Reviewers' comments:

We thank the reviewers for their thorough evaluation of our manuscript and the suggestions and comments for further improvement. For this revision, we have

- *performed extensive additional experiments (32 new ChIP-seq, 3 new RNA-seq, 2 new WGBS),*
- *recalculated all data sets to include the new biological replicates and revised all figures accordingly,*
- *added 7 new figures (Fig. S1, S3, S4, S7, S11, S14, S15),*
- *added 4 new tables (Table S1, S2, S3, S4),*
- *addressed all of the reviewers' questions and comments and revised the manuscript accordingly.*

Reviewer #1 (Remarks to the Author):

General comment:

Reviewer 1: The authors describe the human cardiac myocyte epigenome at three stages during live (fetal, infant and adulthood) and at one diseased condition of end stage heart in adults. The study is of high interest to the cardiac community however, the manuscript lacks scientific clarity, reads very polished, holds wrong estimations and statements.

Response: For the revised version of the manuscript, we have performed 37 additional next generation sequencing experiments to get the maximum number of histone marks, RNA and DNA methylation results from all available samples. For 9 tissue samples, we have obtained RNA, DNA methylation and ≥ 6 histone marks. These additional experiments further strengthen the main findings of the manuscript.

A) Major general points:

A.1) Biological replicate number and characteristics

Reviewer 1: The author studied 6 fetal, 4 infants, 5 non-failing and 6 failing human cardiac samples. The undertaken analysis of WGBS, RNA-seq, histone-ChIP-seq and 5hmC were performed at different samples of different cases. This fact is only given in the Supplemental Table S1, not as a part of the general manuscript. The manuscript is written such that one assumes that all experiments were performed on all samples.

Here is an example given for the fetal samples, which holds true also for all other stages: two cases had WGBS, two different cases had histone-ChIP-seq (H3K9me3 only one sample), one different case had 5hmC; and 4 cases not overlapping with the WGBS cases had RNA-seq.

The study presented in the manuscript has a clear inter-individual bias, as data that are connected and analyzed in relationship to each other (for example RNA-seq

related to WGBS) are generated from samples of different individuals and moreover, the sample number of two does not allow any statistical evaluation.

The manuscript should include an overview explaining which experiments were performed from which case in addition to the supplemental table S1.

In summary, the cross-sample analysis of related data in combination with small sample numbers of each experiment provides a statistical problem. Here the comparison of selected data with previously obtained data in non-cardiomyocyte isolated samples might be helpful to bring confidence. Publications suitable for this are cited in the manuscript and moreover a WGBS tracks of a human young as well as an adult cardiac sample was generated by the Epigenome Consortium.

Response: To provide more information we divided Table S1 into two separate tables, showing patient (new Table S1) and sequencing (new Table S2) information. As suggested, we completed the biological replicates to $n \geq 3$ for all experiments, except for 5hmC-seq. For 5hmC-seq we used a pooled sample to cope with the high DNA input requirements. Thus for this revision, we generated 32 new ChIP-seq, 3 new RNA-seq and 2 new WGBS data sets. In total our study now includes 84 ChIP-seq, 14 RNA-seq, 16 WGBS and 4 5hmC-seq data sets (new Table S2).

As displayed in new Table S2 we have complete data sets now for three adult non-failing and three adult failing, two infantile and one fetal sample. For the remaining samples, we tried to perform as many experiments per sample as possible. However, due to the limited amount of tissue especially from fetal and infantile samples this was not always possible (i.e. fetal approx. 100 mg of LV). We describe this issue in detail in the revised manuscript.

To further address the selection of tissues and potential inter-individual bias, we added three supplementary figures showing genome-wide correlations for all data sets generated for this study (Fig. S3, S4, S5). These analyses show highly correlating data for biological replicates from each age group indicating that inter-individual differences are smaller than differences between fetal, infantile and adult stages.

For comparison with data generated by the Encode and Roadmap consortia, we added a new supplementary Fig. S7. This figure highlights that WGBS data of cardiac myocytes and heart tissue show a substantial overlap. 80% of the VISTA heart enhancers were confirmed by our data. The remaining VISTA enhancers were associated with non-cardiac myocyte genes and functions.

Reviewer 1: Case descriptions for all stages given in table S1 would clearly benefit from more details of the phenotype, in particular as the analysis was performed across samples and their comparability is essential. For example, in four fetal cases the age is given in the range of 18-21 weeks, moreover as abort reason it is just stated “non-cardiac”. A clinical description would be helpful as in case of insufficient hemodynamic supply, the heart could be affected even if the cardiac structure is grossly normal. In case of failing hearts, where these dilated cardiomyopathies, hypertrophic cardiomyopathies or any else?

Response: In the revised version, we added the requested additional clinical information (new Table S1). In addition - to comply with ethical requirements - we had to

unassign patient information and generated data. This separation was necessary to avoid the possibility of patient identification. In case of fetal samples, we provide individual ages. Unfortunately, we are not able to provide further details for fetal tissues. These tissues were obtained from abortions due to non-medical reasons.

A.2) Cell type-specific analysis versus tissue

Reviewer 1: In the introduction, the authors write that data produced in the past by considering total biopsies and not selected cardiomyocytes are misleading and simply speaking bad data. I agree that isolated cardiomyocytes are better, but, data generated at different levels (RNA-seq, ChIP-seq, WGBS) as in previous total samples studied by others were frequently generated from one sample, and moreover these studies considered a statistically significant number of samples.

Response: A comparison of data generated in this study and previously published data sets by the Encode and Roadmap consortia show similar results in large parts of the genome. Especially the annotation of chromatin states in cardiac myocyte genes was practically identical. However, to analyze the stability of DNA methylation in cardiac myocytes in diseased hearts cell type specificity was pivotal. This strategy was the only solution to discriminate between changes of DNA methylation due to altered cell composition or dynamic changes in a specific cell type.

A.3) Data analysis strategy

Reviewer 1: The description and presentation of the computational analysis is crossly insufficient. There are no quality plots or quality data provided in the supplement. Not even a simple PCA plot showing the relationship of the different samples to each other is provided. The presented inter-individual analysis as described above is essentially depending on a clear co-clustering of related samples. The description of the computational analysis is as bad, as not even the methylation levels used to define the used categories four UMR, LMR, PMR and FMR can be found. At least I searched hard and did not find them. It is insufficient to reference two articles from 2011 and 2013 for this.

Moreover, different methylation categorizations are currently in use, e.g. the Meissner group (Tsankov et al. Nature 2015) uses just three categories, namely, unmethylated regions (UMR, where $0 \leq \text{UMR} \leq 10\%$ methylation), intermediate methylated region (IMR, where $10\% \leq \text{IMR} \leq 60\%$ methylation), highly methylated region (HMR, where $60\% \leq \text{HMR} \leq 100\%$). A rationale, for the applied categories would be helpful.

Response: To avoid only citing previously published methods, we provide a detailed description of DNA methylation-guided segmentation of the genome and all bioinformatic analysis in the revised version of the manuscript. The rationale for the applied categories is explained in detail in the revised manuscript. Partially methylated domains have been associated with inactive chromatin by several groups. UMRs are large regions (>30 CpGs) with low CpG methylation. The majority of these regions overlaps genic regions, while LMRs represent cis-regulatory regions (new Fig. S11).

The revised manuscript contains correlation plots of ChIP-seq (Fig. S3), RNA-seq (Fig. S4), WGBS (Fig. S5) data for all biological replicates.

Reviewer 1: Moreover, why not using DMRs? For time-series data, as in the manuscript, the application of DMRs is a common approach (Schultz et al. Nature 2015; Ziller et al. Nature 2013).

DMRs provide a better quantification of methylation changes between different samples than used qualitative categories, and therefore would be a better option for relating alterations of DNA methylation and Histone marks (quantitative ChIP-seq data) to each other, which is an essential part of the study.

Response: The referenced studies (Schultz et al. Nature 2015; Ziller et al. Nature 2013) calculate significant DMRs without incorporation of biological replicates. We used Metilene⁶, which allows a replicate-based analysis of DMRs. In the revised version of the manuscript, we describe this matter as part of the results section. The incorporation of replicates is as suggested by Reviewer 1 of great scientific importance.

A.4) Pre- versus postnatal

Reviewer 1: The study does not take any advantage of having three time points throughout live, but this sample collection could be the strength of the study. Why are methylation information in Fig7 summarized for all samples? What are the differences between neighbored stages? How does DNA methylation changeover time.

5) Moreover, the authors mainly compare fetal to adult samples, without consideration of the intermit of infant samples. This is scientifically not sound, a comparison between 18-21 weeks old fetal cardiac samples to 50-60y old adult samples has little information. The cardiovascular fetal circulation is totally different from an infant circulation and a comparison between these would be very interesting as it addresses this hemodynamic adaptation. I skip any explanations here of the circulatory differences as this is common knowledge. Comparing infant to aged adult samples is also very informative for the study of cardiac aging.

However, the majority of results provided compare fetal to adult cases and therefore mix hemodynamic and aging processes, this is of very little scientific value. Thus, a large part of the result section should need to be reanalyzed to correct this and the manuscript to a large extend needs to be re-written. The project summary might even be very different.

Response: For the revised manuscript we performed the suggested analysis (new Fig. S14). This analysis clearly shows that dynamic DNA methylation is a continuum. This implication was also confirmed using a principal component analysis (Fig. S15). We are aware and discuss in the revised manuscript that this finding is in contrast to results obtained from pre- and postnatal mouse tissues deposited on a preprint server¹.

A.5) Meaning of "Development"

4) The author misuse the term “development”. This term should be applied for the comparison between fetal and infant samples. The term “aging” should be used for the comparison between infant and adult stages. This needs to be corrected throughout the manuscript.

Response: In order to be more precise, we have now separated the prenatal development from the postnatal period. Although the first period covers the time between fetal weeks 17-23 and infant ages 2-12 month, we use the term "prenatal development" for comparison between these time points. For comparison of infantile and adult samples, we suggest to call this period "postnatal maturation". Our adult non-failing samples were obtained from patients between 46 and 60 years of age and did not reach the average life expectancy in the Western world, i.e. over 80 years. Thus instead of "aging", we suggest to use "postnatal maturation" or "maturation" for this study. The revised manuscript and all figures were corrected accordingly.

B. Selected major specific point:

1) Figure 1c, shows the proportion of cardiac myocyte nuclei in fetal (Fe, n=3), infantile (I, n=5), adult non-failing (NF, n=5) and adult failing (F, n=5) LV tissue. In the figure are five data points for each of the samples type shown. How can this be, if Fe has only a n=3 as stated in the figure legend, moreover, in table S1 are only four infantile cases/samples described, how can five infantile samples be part of this figure and its legend?

Response: In the revised table S1 we describe all assessed samples in this study. The former table S1 only listed samples with available sequencing data. We corrected the n-number in Fig. 1c shows data of three fetal and five infantile, non-failing and failing hearts.

2) In the manuscript, one central claim, presented twice is that: “Remarkably, 33% of the cardiac myocyte CpG methylome was shaped after birth between infant and adult stage.”

This sentence is totally incorrect at least considering the data presented in the manuscript. Correct is that 33% of the dynamic LMRs differ between infant and adult stages; dynamic LMRs represent only 15% of all LMRs. Moreover, the cardiac methylome consists of UMRs, LMRs, PMRs and FMRs. Therefore, correct is that below 0.1% of the cardiac myocyte CpG methylome is shaped after birth between infant and adult stages, and this is not surprisingly.

Response: In the revised manuscript, we added a new analysis showing pre- versus postnatal dynamics in DNA methylation. We therefore modified the results and discussion and do not mention the “33%” anymore. We agree that this percentage may lead to misinterpretation. For your information, taken all parts of the methylome together 0.4% of the genome is altered postnatally.

C. Selected minor points:

C.1) Figures could be homogenized, for example in Fig 2c, genes are given for gUMR whereas in the comparable analysis of LMRs in Fig 4a, LMRs regions are given.

Response: As introduced in the results, discussion and method section gUMRs represent UMRs overlapping genic regions including the transcription start site. Therefore, gUMRs are directly linked to specific genes. Therefore numbers and names are given. The definition of gUMRs is explained in the methods section in great detail.

In contrast, LMRs are cis-regulatory sites and interacting genes are in general not known.

C.2) Statistical values are frequently missing, it is just stated that something would be significant, see for example Fig 4a.

Response: In the revised manuscript we describe the replicate-based DMR annotation performed in this study in detail. To highlight this fact throughout the manuscript we introduce the terms DM-LMR and DM-gUMR. We added further technical details to the methods sections and checked, that statistical values are given, if stated.

Reviewer #2 (Remarks to the Author):

This manuscript by Gilsbach et al. maps the epigenome (7 histone marks), methylome (whole-genome bisulfite sequencing) and transcriptome (RNA-Seq) of human cardiac myocytes during development of the heart from fetal to adult. They refined methods for purifying cardiomyocyte nuclei from tissue by flow cytometry and analyzed cardiac myocytes at 3 time-points: fetal, infantile and adult along with comparing between adult non-failing and terminally failing hearts. They first segmented the genome according to its methylation status and then by comparing the nearest-gene annotation and epigenetic signature determined that low-methylated regions are putative enhancers whereas un-methylated regions are present in gene bodies and CpG islands. They found that methylation status and histone marks at gene bodies are correlated with gene expression for the subset of genes that are developmentally regulated during myocyte maturation. In their study of pathologically important genes for heart failure however they found that while active histone marks could to some degree explain the change in gene expression, the same was not true for inactive marks and CpG methylation. They then repeated this analysis for the low-methylated regions (putative enhancers) finding similar interplay between changing methylation, histone signature and gene expression during development. They again could not find any pattern for disease-associated genes. They also found that cardiac disease associated genetic polymorphisms were enriched at these low-methylated regions.

The authors have carried out a substantial body of work and generated datasets that will be valuable to those studying heart development, heart failure, or cardiac gene regulation. The methods for isolation of cardiomyocyte nuclei will also be valuable. However, the major conclusion of the paper that methylation status and the epigenome are closely linked to changes in gene expression during CM maturation, is not novel and has been shown in other developmental systems. They also studied changes during heart failure but did not have major novel findings.

A) Major concerns:

A.1-2: Novelty

1. Novelty, as noted above.

2. The authors reported a similar study of murine cardiomyocytes in Nat. Comms 2014. There is little comparison of the results of this study of human cardiomyocytes to the prior work on murine cardiomyocytes. Were the major findings in each system consistent? What is the conservation of epigenetic marks, especially DNA methylation? Aside from providing information on human cardiomyocytes, what advances did this study make over the 2014 study?

Response: The data generated in this study provides several new insights.

1) We present the first nuclear marker for prenatal and postnatal cardiac myocytes (PLN).

2) The cell type-specific analysis presented in this study shows for the first time that pathological gene expression in cardiac myocytes is orchestrated by changes in active histone marks but neither alterations in repressive histone marks nor DNA methylation.

3) The 2014 study presented in *Nature Communications*¹⁰ is less comprehensive and contains a substantially lower number of findings. In fact, only Fig. 1c, f and 2a, b show analyses which partly overlap with our study in postnatal mouse cardiac myocytes. The 2014 study lacks fetal data and only provides WGBS for diseased cardiac myocytes (but no ChIP-seq or RNA-seq data for failing myocytes).

4) In those cases where we have data from mouse and men, the data are strikingly consistent. Genic UMRs were observed in the same set of genes and even the number of LMRs is comparable. LMR and gUMR prediction was not part of the 2014 study and it is beyond the scope of this manuscript to show a direct comparison of mouse and men.

5) We provide a comprehensive annotation of cardiac myocyte enhancers which are strongly enriched for disease-associated genetic variants.

A.3: Replicates and reproducibility

3. The authors report having done replicates, but do not provide information about the reproducibility of the data. How were replicates combined?

Response: For the revised version of the manuscript, we have analysed correlation of data obtained for all individual biological replicates. These data show the high correlation of RNA-seq, ChIP-seq and WGBS data generated for this study (new Fig. S3, S4, S5). As stated in the revised methods part, replicate-based statistical tests were performed for differential gene expression and DNA methylation. We display merged data from all biological replicates, since showing all individual replicates is practically impossible. In total, we have generated 119 next generation sequencing data sets. We added the information that we display merged data to the legends and the methods section.

A.4: Replicates and reproducibility

4. The authors should provide more data about the human myocardial samples, especially the failing heart samples. There is considerable heterogeneity in human heart failure, and many causes of heart failure. The cardiac diagnoses should be provided. The authors should consider the possibility that negative findings in heart failure were due to heterogeneity and lack of sufficient statistical power. Was there more intersample heterogeneity in the failing samples than the other groups? How would this impact the analysis and the ability to detect differences? Along these lines, on page 8 the authors show that gene expression in disease is not correlated with DNA methylation or histone repressive marks and therefore conclude that these marks are stable. However, lack of correlation cannot be equated to stability.

Response: In the revised version of the manuscript we provide additional patient information in Table S1. We did not observe increased heterogeneity in DNA methylation data obtained in failing as compared to fetal, infantile or non-failing samples. In fact, at loci with sufficient coverage DNA methylation of biological replicates obtained from non-failing and failing samples values are genome wide indistinguishable. This observation is supported by the high correlation of WGBS

data from different biological replicates. For all failing hearts the correlation is 0.99 (Fig. S5).

In case of gene expression, we clearly see a higher heterogeneity (Fig. S4) in data obtained from adult samples as compared to experimental disease models (i.e. mouse transverse aortic constriction). To be able to detect differences we included only failing hearts with terminal chronic heart failure and non-failing samples with no signs of hypertrophic heart disease. We analyzed DNA methylation in an unbiased genome-wide fashion as well as at loci of differentially regulated genes. In addition, to be sure not to miss dynamic changes in DNA methylation we visually inspected all loci known to be related to heart disease.

The data generated in this study clearly show that heart failure does not induce disease-relevant changes in DNA methylation. However, we cannot exclude that genetic variants affect epigenetic signatures, which are important for the development or progression of heart disease. Future studies assessing large cohorts are necessary to unravel this interplay.

A.5-7: Enhancer prediction

5. The authors started by segmenting the data according to methylation status. They did most of their analysis on gene-unmethylated regions (which are basically gene bodies) and low-methylated regions (putative enhancers). The advantage of this approach over chromatin-based segmentation to divide the genome into promoters, enhancers, inactive and other regulatory regions is not clear. Maybe the authors can show some type of clustering analysis to illustrate why this is a superior approach. Does adding DNA methylation to histone data in ChromHMM or similar modeling substantially influence segmentation into functional elements?

Response: DNA methylation-based annotation is not superior to histone code-based annotation of regulatory elements. We focused on DNA methylation-guided segmentation of the genome, since this strategy identified genomic elements with characteristic DNA methylation patterns. The precise identification of these elements (LMRs, UMRs, PMRs) was the basis for our differential methylation analysis. In the revised manuscript, we used ChromHMM to annotate functional elements in the genome. Comparing the annotation of both methods showed compatible results (Fig. S11). LMRs were enriched for enhancers, UMRs for promoters and PMRs for silent chromatin (Fig. S11b). Unfortunately, ChromHMM does not accept DNA methylation data. For sure, future algorithms integrating all different epigenetic data will strengthen the annotation of regulatory elements. A promising attempt for tissue data was recently published¹.

6. The authors should compare the putative enhancer regions identified in their study to those identified in other studies of human heart enhancers (e.g. May et al., Nat Genet; Dickett et al, Nat Comms). What is the sensitivity and specificity of detecting regions with transgene-proven heart enhancer activity (e.g., VistaEnhancer database)? Does the methylation data improve detection of enhancers?

Response: In the revised manuscript we added a new figure comparing the data generated for this study with data generated by Encode and predicted as well transgene-

proven VISTA heart enhancers⁵ (Fig. S7). This comparison shows a strong overlap of cardiac myocyte enhancers with VISTA heart enhancers (Fig. S7g). 82% of predicted VISTA enhancers were also detectable in cardiac myocytes. However, 18% of VISTA heart enhancers regions were not overlapping with enhancers detected in cardiac myocytes (Fig. S7g, blue part of pie chart). These non-overlapping enhancer regions were associated with genes essential for endothelial cell and fibroblast function (Fig. S7h). Transgene-proven VISTA heart enhancers showed an even greater overlap with cardiac myocyte enhancers (Fig. S7i). 90% of in vivo proven VISTA heart enhancers overlapped with cardiac myocyte enhancers predicted by ChromHMM and 80% contained LMRs or UMRs (promoter regions) (Fig. S7i).

It should be noted, that transgene-proven VISTA heart enhancers lacking LMRs in cardiac myocytes contained only weak enhancers. Such weak enhancers are normally associated with sub-threshold (50%) loss of DNA methylation. Likely these enhancers are active either transiently or in a cardiac myocyte subpopulation.

We think that CpG methylation as well as histone modification-guided annotation of regulatory elements are compatible and important. For this project the CpG methylation-guided segmentation was of special importance, since it was the basis for the identification of differentially methylated regions. We added a new paragraph to the discussion focusing on DNA methylation and chromatin state-based annotation of enhancers.

7. The authors propose that low methylated regions (LMRs) are cis-regulatory elements. However, only half of the LMRs are marked by H3K27ac. What about the other half? Are H3K27ac+ LMRs associated with greater gene expression than H3K27ac- LMRs? What fraction of H3K27ac regions have LMRs?

Response: For the revised version of the manuscript, we annotated enhancers using ChromHMM. According to ChromHMM more than 70% of LMRs have an enhancer-like chromatin state (Fig. S11d). This state was characterized by an enrichment of H3K4me1 and H3K27ac combined with a lack of H3K4me3 and therefore allows a more precise annotation of potential enhancers. As expected, LMRs with enhancer chromatin state are associated with higher gene expression as compared to LMRs with silent chromatin state (Fig. S11e).

A.5-7: RNA-seq method

8. For transcriptome analysis the authors use cufflinks suite. Though cufflinks is widely used, it is known to underestimate differentially expressed transcripts. Maybe the authors can redo transcriptomic analysis using a second method (e.g RSEM, EBSeq if the authors are interested in transcript level analysis or DESeq2 if gene-level) and show the agreement between methods to ensure that any differential expression analysis is not driven by the choice of methods. Also in the methods section on page 20, line 7: the authors talk about selecting the main cardiac myocyte isoform of each gene. It is not clear how this was determined. If this was based on expression it can be problematic as isoforms might be differentially expressed during development. Due to these concerns it seems best to carry out gene-level differential expression analysis with a count-based method.

Response: For the revised version of the manuscript, we used a count-based analysis (HTScount + DESEQ2). As expected the number of differentially expressed genes identified by this method is slightly higher as compared to Cuffdiff. Most importantly, the choice of differential gene expression method had no impact on the main findings of the manuscript. For these analysis we used all isoforms annotated by the UCSC. The Tuxedo suite was used for the identification of the major transcription start (TSS) and end sites (TES) of the main isoform and gene expression values (FPKM). Remarkably, we did not observe differential TSS and TES usage in the assessed samples. We added this information to the method section. TES and TSS of the main isoform was for visualization and analysis of epigenetic data. We clearly describe this strategy in the revised methods section.

B) Minor concerns:

B.1: On page 6, line 19; its not clear whether the genes are expressed in prenatal, postnatal cardiomyocytes or either.

Response: In the revised version, we added a Venn diagram illustrating the overlap of genes detected (> 1 FPKM) at the assessed stages (new Fig. S4b).

B.2: On page 6, line 19; the authors should mention some quantitative statistic detailing the level of correlation between CpG methylation and gene expression.

Response: Due to the skewness of gene expression levels, only ranking analysis reveals a correlation between genic CpG methylation and gene expression, only hundreds of genes show very high expression levels as compared to thousands of genes with very low or undetectable expression. This distribution is problematic for quantitative statistics. We also should consider the existence of approx. 170 developmental genes overlapping CpG methylation valleys¹¹ (Fig. 2B, group II). These genes show low CpG methylation values and are not expressed in cardiac myocytes at the assessed stages, but mostly are known to be expressed at earlier stages (i.e PITX2, ISL1, NANOG). This highlights that CpG methylation functions as the epigenetic memory of the cell¹². These issues argue against a quantitative correlations of gene expression and genic CpG methylation.

B.3: On page 7 line 1; the authors talk about 2 groups of genes associated with different signature and enriched for different GO terms. The list of genes in both group I and group II can be informative for the field and should be detailed in a supplementary table along with at least the top 10 Go terms enriched

Response: In the revised manuscript we added a new Supplementary Table S4 to display the requested data.

B.4: On page 8, line 17; The exact methodology for ranking the genes should be detailed in the methods section.

Response: We added a new chapter to the methods section explaining this methodology.

B.5: On page 9, line 2; The list of LMRs with decreasing or increasing CpG methylation during development could be informative for the field and should be detailed in a supplementary table.

Response: We added a new supplementary table S4 with the requested information.

B.6: On page 19, line 27; Usually for ChIP-Seq, it is preferred to keep only those reads that map uniquely to the reference genome. In Bowtie this can be done with -m 1 option. However Bowtie2 doesn't have this feature and requires additional processing steps downstream. Did the authors do additional processing?

Response: We did not do additional processing of Bowtie2 mapped reads despite of removing duplicates. We chose the default Bowtie2 option that reads, which do not uniquely map to the genome were assigned to the first best possible alignment. This unbiased approach is especially important for the analysis of repetitive regions. Choosing Bowtie1 with option -m would result in low coverage of repetitive genome regions. To increase the number of uniquely mapping reads we sequenced in paired-end mode.

B.7: On page 19, lines 30-32 are repeated

Response: We corrected this mistake.

Reviewer #3 (Remarks to the Author):

Gilsbach et al present a paper describing the epigenetic status and transcription factor networks at various stages of human cardiac pre-natal and post-natal development and in chronic heart failure.

The aim of this study is of great interest and novelty in the field and it has the advantage of studying purified cardiomyocytes (nuclei) obviating to the confounding data obtained by whole tissue studies.

The authors' conclude that during differentiation of adult cardiomyocytes, the induction of specific genes (i.e. MYH6, RYR2; TNNI3) or repression of others (MYL4, NPPA, TNNI1) depend on the combination of methylation levels at enhancers and gene bodies as well as histone marks while . In contrast, induction of fetal genes in cardiomyocytes of failing hearts is associated to active histone marks but no or little DNA methylation changes. Importantly, the authors have identified SNPs, previously associated to cardiac disease, in enhancers displaying low methylation.

Overall, the paper is well written and the presentation of the data is logical although the reviewer has some suggestions and comments.

A1) Sample characteristics

The authors used samples from four groups of donors with 4-6 donors/group. It would be important to summarize in brief the characteristics of the donors shown in table S1 in the results section. The cohort used for the present study is good, however, it would be important to have a better understating of how the donors were selected and their clinical history including for example history of congenital diseases, cancer, infections, diabetes and any previous or current treatment. The reviewer would like to know the time frames between deaths, tissue harvest and tissue processing from all the samples of each group. This information should be taken in account as a potential technical factor influencing the results.

Response: We added the information how the tissues were processed after death or during surgery to the methods section (first paragraph). We were positively surprised, that the delay between death and tissue harvest did not comprise the integrity of the nuclear RNA and chromatin as compared to tissue snap frozen during interventions. Possibly, the nuclear RNA and the chromatin is well preserved within the nuclear space. However, a low number of freeze thaw cycles is sufficient to degrade chromatin and nuclear RNA.

In the revised manuscript we added a new Table S1 showing detailed patient characteristics.

A2) FACS

It would be useful, for clarity, to spell out in the results (not just the discussion) that the nuclear marker pericentriolar material 1 (PCM-1) enriches for cardiomyocytes' nuclei not any nuclei (Bergmann et al 2010, 2011 and 2015). Indeed, this would help interpreting Figure S1 and S2. Figure S1 is hard to interpret because of the lack of gates or quadrants and the use of a different scale for the axes.

Assuming that isotype controls were used for each sample, the reviewer suggests to add the isotype controls to Fig S2, using same scale axis for each set in conjunction with gates or quadrants. Additionally, the reviewer would like to see a more comprehensive representation of these data: in Fig 1e only the percentage of cardiomyocytes nuclei is shown, it would be useful to see in a graph the summary of the percentages of the other populations in the various groups of patients, not just a representative dot plot. This would be important in view of the abundant PCM1+PLN- cells seen in infants samples (Fig S2) as this might represent a specific set of cardiomyocytes or a specific stage of differentiation. Interestingly, no PCM1-PLN+ population appears in the dot plots shown in Fig. S2. Additionally, data from Bergmann et al 2015 suggests that the number of endothelial and mesenchymal cells in the myocardium increases in adult hearts, thus, the reviewer would like to see whether there is a consistent increase in the PCM1-PLN- population from fetal to adulthood and if these represent endothelial/mesenchymal cardiac cells and/or nuclei.

Response: We added the suggested statement that “PCM-1 is specific for cardiac myocytes in the postnatal heart” to the results section.

We added a new supplementary figure showing FACS diagrams for all assessed stages including isotype controls. All figures were obtained using identical FACS and analysis settings. This analysis shows clearly that PLN staining is evident at all stages, while PCM-1 staining only in adult non-failing hearts. We removed quadrants from figure S2 and show sorting gates. The position of the quadrants was misleading, they splitted populations of nuclei.

Our results perfectly fit to results obtained by Bergman et al. 2015. We see a drop of the fraction cardiac myocyte in the heart from approx. 80% (fetal and first postnatal year) to 30% in adult hearts. This supports the postnatal proliferation and invasion of non-cardiac myocytes including endothelial cells and fibroblasts. Due to technical reasons, we were not able to provide the fraction of fibroblasts and endothelial nuclei in this study.

A3) ChIP differences

The reviewer is intrigued by the comments on page6 lines 8-14 regarding demethylation, however, the data presented in Fig S8 could be presented in a more convincingly way to support the statements. In essence, where and how do the authors prove that UMRs mirror demethylation? Are the authors presuming this based on the induction of a given gene expression or do they have direct evidence? In addition, how do the authors address and explain the potential differences seen at the various stages analysed from a statistical and biological point of view (see for example Fig S8 j, k and i)?

Response: We rephrased the misleading term “To specifically analyze gene body demethylation.” to “To specifically analyze characteristics of genes with low methylated gene bodies.” To gain mechanistic insights from the data presented in this study is not possible. We think that gene expression drives loss of CpG methylation in differentiated cells. In pluripotent cells this is counteracted by the recruitment of DNMT3b⁹, and DNMT3-isoform which is very low expressed in postnatal cardiac myocytes. Further studies are necessary to prove this hypothesis.

We agree that gradual changes in ChIP-seq enrichment exist between different samples. We think that these differences are rather technical and not biological. Therefore, we decided not to perform statistical comparisons based on ChIP-seq data (i.e. regions with sign. different H3K27ac signal). Nevertheless, qualitative comparisons of i.e. H3K27ac enrichment at regions with loss or gain of CpG methylation are informative and are used for example for correlative analysis. This limitation of current ChIP-seq technology led us to a DNA methylation centered analysis, since WGBS is quantitative. In the revised version, we added ChromHMM⁷ annotation tracks. Chromatin state annotation is in our view less affected by ChIP-seq biases.

A4) Scheme

The scheme in Fig 7 is very useful in summarizing the message, the legend could benefit from some clarification as it is unclear whether UMR and LMR that in the brackets refer to the induced genes or repressed genes. Also, the scheme should highlight better the differences between gene body and enhancers methylation. In addition, would it be appropriate to talk about gene induction albeit demethylation is deficient in the failing hearts? Are the levels of genes expressed in these circumstances lower compared to a situation where you have UMRs plus active histone marks? How do you expect the expression to be in the fetal and earlier phases of cardiomyocytes differentiation and function in the same patients?

Response: In the revised version, we changed the labels of LMRs and genic UMRs to enhancer and gene. This highlights the location of the DNA methylation changes depicted in the graph much better. In addition, we rephrased the main summary of the figure in the legend. Gene expression analysis suggests that DNA methylation-modifying genes are downregulated but not absent in post-mitotic adult cardiac myocytes. Therefore, we cannot prove that modulation of DNA methylation is deficient in adult cardiac myocytes. So far, we have no experimental proof that genic UMRs facilitates gene expression. In future studies, we will address this question in vitro cell models.

A4) iPS-cells

It would be intriguing to generate iPS-cells from failing patients and test the epigenetic and transcriptome in relation to these SNPs during various stages of cardiomyocytes differentiation.

Response: The sample size of this study is not sufficient to identify disease-associated SNPs and their impact on epigenetic signatures and gene expression. Future studies assessing large cohorts are necessary to unravel the interplay between epigenetic mechanisms and genetic risk factors in heart disease. The suggested iPS cells will be the final prove for this hypothesis.

A4) Visualization

The authors show original traces (i.e. Fig1f and 3a): can they clarify how these data are displayed? Do they always show the same sample/donor for each group and could they clarify which sample is it? Could they also clarify in the instances where they pool data from all the donors of each group as described in TableS1?

Response: We added supplementary figures showing the correlation of WGBS, RNA-seq and ChIP-seq data generated from biological replicates (Fig. S3, S4, S5). In all main figures we show identically merged data. We added this information to the legends and the methods section. The assignment of samples and data is depicted in the new Supplementary table S2. We also provide a new Supplementary table S1. To comply with the ethical approval of this study, we had to unassign patient and sample IDs.

B) Minor issues

B.1: Page 3, line 26: although this is stated elsewhere, the reviewer suggest to add a statement describing what is changing in the pathological conditions studied (active histone marks) given that no CpG methylation modifications were detected.

Response: We added the suggested statement to the last paragraph of the introduction.

B.2-4: Fig1 e and especially f: the colour used for 'Fe' samples is hardly distinguishable from 'I' and 'NF' Fig S8, it is difficult to distinguish the colours, this could be obviated by increasing the thickness of the lines and/or changing the colours. In the figures, in several instances the lines of the curves are too thin. The font of main and supplemental figures' labelling is in most cases too small and hard to read.

Response: As suggested, we increased the thickness of the lines as suggested and adjusted the colour of the fetal (Fe) sample throughout the manuscript. We increased all font size not complying with the journals style guide.

References:

- 1 He, Y. *et al.* Spatiotemporal DNA Methylome Dynamics of the Developing Mammalian Fetus. *BioRxiv*, doi:10.1101/166744 (2017).
- 2 Gorkin, D. *et al.* Systematic mapping of chromatin state landscapes during mouse development. *BioRxiv*, doi:10.1101/166652 (2017).
- 3 Kundaje, A. *et al.* Integrative analysis of 111 reference human epigenomes. *Nature* **518**, 317-330, doi:10.1038/nature14248 (2015).
- 4 Schultz, M. D. *et al.* Human body epigenome maps reveal noncanonical DNA methylation variation. *Nature* **523**, 212-216, doi:10.1038/nature14465 (2015).
- 5 Dickel, D. E. *et al.* Genome-wide compendium and functional assessment of in vivo heart enhancers. *Nature communications* **7**, 12923 (2016).

- 6 Juhling, F. *et al.* metilene: fast and sensitive calling of differentially methylated regions from bisulfite sequencing data. *Genome research* **26**, 256-262, doi:10.1101/gr.196394.115 (2016).
- 7 Ernst, J. *et al.* Genome-scale high-resolution mapping of activating and repressive nucleotides in regulatory regions. *Nature biotechnology* **34**, 1180-1190, doi:10.1038/nbt.3678 (2016).
- 8 Clark, S. J. *et al.* Joint Profiling Of Chromatin Accessibility, DNA Methylation And Transcription In Single Cells. *bioRxiv*, doi:10.1101/138685 (2017).
- 9 Baubec, T. *et al.* Genomic profiling of DNA methyltransferases reveals a role for DNMT3B in genic methylation. *Nature* **520**, 243-247, doi:10.1038/nature14176 (2015).
- 10 Gilsbach, R. *et al.* Dynamic DNA methylation orchestrates cardiomyocyte development, maturation and disease. *Nature communications* **5**, 5288, doi:10.1038/ncomms6288 (2014).
- 11 Xie, W. *et al.* Epigenomic analysis of multilineage differentiation of human embryonic stem cells. *Cell* **153**, 1134-1148, doi:10.1016/j.cell.2013.04.022 (2013).
- 12 Hon, G. C. *et al.* Epigenetic memory at embryonic enhancers identified in DNA methylation maps from adult mouse tissues. *Nature genetics* **45**, 1198-1206, doi:10.1038/ng.2746 (2013).

Reviewers' comments:

Reviewer #1 (Remarks to the Author):

The manuscript by Gilsbach et al. has greatly improved during the revision. The authors performed extensive additional experiments and recalculated all data sets to include the new biological replicates ($n \geq 3$ for all experiments except 5hmC-seq). As the analyses have been performed at different samples of different cases, they analyzed the correlation for all the generated WGBS, RNA-seq, ChIP-seq data sets which shows highly correlated data for the replicates of each group indicating that inter-individual differences are smaller than differences between the stages during fetal development and postnatal maturation. In addition, they compared methylation changes during development of fetal and maturation of infantile cardiac myocytes (i.e., instead of comparing fetal to adult samples) and found that dynamic CpG methylation is a continuum, which is surprisingly in contrast to results obtained from pre- and postnatal mouse tissues. Moreover, the authors substantially extended the methods sections which is very helpful to fully understand the certain analysis steps. Overall, the manuscript is now basically sound and its merits are compelling. However, there are some points which need to be addressed by the authors.

1. Based on the additional clinical information provided in the new Table S1, it turns out that two out of five infant heart samples show pathological cardiac characteristic. One of these two samples (i.e., I4 if the sample name in Table S2 is correlated to the patient ID in Table S1) is part of the replicates for RNA-seq and all ChIP-seq analyses. The related heart showed a minimal aortic and tricuspid valve insufficiency and developmental retardation. This should to be further analyzed and discussed in detail. On average, there is a high RNA-seq and ChIP-seq data correlation between the three biological replicates of the infant stages; however, for single features and in direct comparison to other stages this might be different. For example, the Spearman correlation for H3K4me3 (Fig. S3a) is 0.87 in I4 vs. I1 and 0.85 in I4 vs. I3 (infant vs. infant) but 0.9 in I4 vs. F4, 0.92 in I4 vs. F3 (infant vs. failing).

2. The authors show that there are distinct differences in DNA methylation and chromatin state between tissue and purified cardiac myocyte nuclei. Based on the analysis and moreover, by visual inspection of loci containing myocyte-specific and non-cardiac myocyte genes (Fig. S7), this statement holds true for the gene bodies. But what about the promoter regions, which are finally more important for driving the expression? For example, for MYH7 (Fig. S7b) there is a clear difference over the gene body in CpG methylation between fetal CM and fetal heart. Nevertheless, the methylation level at the (distal) promoter region looks more similar (at least by visual inspection).

3. In order to be more precise, the author should say in the manuscript that cardiac tissue was analyzed at three stages (fetal, infant and adulthood) and at one disease condition of end stage heart in adults instead of "at four stages" (e.g., see line 113 or 119 in the manuscript).

4. Left ventricular heart tissue was obtained from fetal between 16 and 23 weeks and not between 17-23 week (see line 114 in the manuscript).

5. Again (see comment B.1), Figure 3c shows the proportion of cardiac myocyte nuclei in fetal ($n=3$), infantile ($n=5$), adult non-failing ($n=5$) and adult failing ($n=5$) LV tissue given. How can it be that for fetal heart the given number of samples and data points does not fit to the given number of samples in Table S1 and S2 (i.e., 3 versus 8)?

6. Figure 3d shows the distribution of cardiac myocyte ploidy in the different LV tissues. The given number of samples does not fit to the given number of samples in Table S1 and S2 (i.e., fetal 3 versus 8, infantile 6 versus 5, failing 4 versus 5)?

7. In Figure S1 is the plot for one infantile and one non-failing heart labeled with some letter ('h').

Moreover, which samples are plotted? The sample IDs should be provided.

8. In Figure S4 is the legend for S4a and S4b mixed. In addition, the Venn diagram contains the number of genes with ≥ 1 FPKM in fetal, infant, adult non-failing or adult failing cardiac myocytes. Is the FPKM threshold on average over all replicates of a group or in at least one replicate?

9. In Figure S9 is the labeling for stable PMRs not correct (i.e., should be 'b' instead of 'f').

Reviewer #2 (Remarks to the Author):

Re: Gilsbach et al., Distinct epigenetic programs regulate cardiac myocyte development and disease in the human heart in vivo

In this revised manuscript, the authors were very responsive to prior the prior reviews. They authors added extensive additional data and analyses. In particular, these revisions went a long way to address the common question raised by all reviewers about data reproducibility and the effect of using different tissue samples.

This is a large body of work that will be highly useful for those interested in epigenetic regulation in development, aging, and disease, particularly within cardiomyocytes.

A few remaining concerns:

1. The authors now provide additional information about the tissue samples. The identifiers of the tissue samples are not consistent so that it is not possible to link the data tracks to the individual tissue sample descriptions.

2. Fig. S3 shows the correlation between samples for histone ChIP-seq data.

- are these correlations made genome-wide, or at called peaks?

- the clustering analysis shows that samples do not cluster by group. This is concerning for intra-group heterogeneity being greater than inter-group differences.

3. The comparison to the heart enhancer compendium of Dickel et al (Fig. S7g) should be presented so that one can see the relative number of regions unique to the predictions of this study or of the Dickel study, rather than only the regions in the intersection.

How do the methylation-predicted enhancers compare to ChromHMM in a comparable analysis?

4. The discussion states, "a triple antibody labeling strategy was necessary and sufficient to establish a sorting strategy for cardiac myocyte nuclei from pre- and postnatal human hearts." However, the data seems to suggest that individual labeling with a single antibody such as PLN would be sufficient to sort the cardiac myocyte nuclei.

Reviewer #3 (Remarks to the Author):

The authors have provided a wealth of additional material that allows the readers to interpret the study thoroughly. This includes new data, updated data presentation and methodological information. All of this makes it a solid study which will be useful to the scientific community.

Page 5, line 175:were involved in cardiac muscle...

Response to Reviewers' comments:

Dear Reviewers, thank you very much for reviewing our manuscript. Below you find the responses to your comments. For the revised version we generated and revised the Figures 1c, d and the Supplementary Figures S11e, S7g-j to address the raised concerns. We describe these new results in the revised version of the manuscript and integrated the suggested changes. The revised manuscript is more concise as compared to the initial version to comply with the formal requirements of Nature Communications.

Reviewer #1 (Remarks to the Author):

The manuscript by Gilsbach et al. has greatly improved during the revision. The authors performed extensive additional experiments and recalculated all data sets to include the new biological replicates ($n \geq 3$ for all experiments except 5hmC-seq). As the analyses have been performed at different samples of different cases, they analyzed the correlation for all the generated WGBS, RNA-seq, ChIP-seq data sets which shows highly correlated data for the replicates of each group indicating that inter-individual differences are smaller than differences between the stages during fetal development and postnatal maturation. In addition, they compared methylation changes during development of fetal and maturation of infantile cardiac myocytes (i.e., instead of comparing fetal to adult samples) and found that dynamic CpG methylation is a continuum, which is surprisingly in contrast to results obtained from pre- and postnatal mouse tissues. Moreover, the authors substantially extended the methods sections which is very helpful to fully understand the certain analysis steps. Overall, the manuscript is now basically sound and its merits are compelling. However, there are some points which need to be addressed by the authors.

1. Based on the additional clinical information provided in the new Table S1, it turns out that two out of five infant heart samples show pathological cardiac characteristic. One of these two samples (i.e., I4 if the sample name in Table S2 is correlated to the patient ID in Table S1) is part of the replicates for RNA-seq and all ChIP-seq analyses. The related heart showed a minimal aortic and tricuspid valve insufficiency and developmental retardation. This should to be further analyzed and discussed in detail. On average, there is a high RNA-seq and ChIP-seq data correlation between the three biological replicates of the infant stages; however, for single features and in direct comparison to other stages this might be different. For example, the Spearman correlation for H3K4me3 (Fig. S3a) is 0.87 in I4 vs. I1 and 0.85 in I4 vs. I3 (infant vs. infant) but 0.9 in I4 vs. F4, 0.92 in I4 vs. F3 (infant vs. failing).

Response: *In order to comply with the regulations of the ethics committees and data banks, we have to unassign patient information and genomic sequence information. Therefore, sample names and patient IDs in this manuscript are not linked. The same holds true for the order of the samples listed in Table S1 and S2. This information is now mentioned in the methods section.*

In the revised version of the manuscript we added further information supporting suitability of the infant heart samples for these epigenomic analyses. According to expert medical and cardiology advice, the minimal tricuspid and aortic valve insufficiency of patient 12 was not hemodynamically relevant for the heart (Suppl. Table S1). The same applies to patient 13 with a partial ductus Botalli. Macroscopic and microscopic inspection of these hearts did not identify signs of pathological remodeling. Furthermore, we did not identify pathological gene expression (i.e. increased Nppa, Nppb, Acta1, Ankrd1 expression) in these hearts.

Sequencing data obtained from these hearts and the correlation coefficient for all assessed epigenetic marks did not differ from the remaining samples. Furthermore, assignment of the H3K4me3 is dynamic at promoters of many genes which are differentially regulated in disease and development, but H3K4me3 marks also CpG islands. CpG island promoters are in several cases H3K4me3 positive irrespective of their transcriptional activity. This pattern is a hallmark of paused promoters. On a genome-wide scale, CpG islands with rather stable enrichment of H3K4me3 outnumber regions with dynamic H3K4me3 enrichment. Therefore, the high correlation between data from different patients (Suppl. Fig. 3a) shows that the experimental bias between the samples is very low.

2. The authors show that there are distinct differences in DNA methylation and chromatin state between tissue and purified cardiac myocyte nuclei. Based on the analysis and moreover, by visual inspection of loci containing myocyte-specific and non-cardiac myocyte genes (Fig. S7), this statement holds true for the gene bodies. But what about the promoter regions, which are finally more important for driving the expression? For example, for MYH7 (Fig. S7b) there is a clear difference over the gene body in CpG methylation between fetal CM and fetal heart. Nevertheless, the methylation level at the (distal) promoter region looks more similar (at least by visual inspection).

Response: *We agree that gene bodies harbor the largest regions with differential methylation between heart and cardiac myocytes. In addition, Suppl. Fig. S7b illustrates that enhancer regions identified by ChromHMM upstream of MYH6 and MYH7 are also higher methylated in heart tissue as compared to cardiac myocytes. To illustrate differential CpG methylation between heart tissue and cardiac myocytes in genes and flanking regions we integrated the new supplementary figure S7g. This figure underlines that promoters and flanking regions of highly expressed cardiac myocyte genes show higher CpG methylation levels in tissue as compared to cardiac myocytes. It should be noted that CpG methylation values of more than 50% of cardiac myocyte-specific LMRs are in average 20% higher methylated in heart tissue as compared to cardiac myocytes.*

3. In order to be more precise, the author should say in the manuscript that cardiac tissue was analyzed at three stages (fetal, infant and adulthood) and at one disease condition of end stage

heart in adults instead of “at four stages” (e.g., see line 113 or 119 in the manuscript).

Response: *We made the suggested change at line 113. At line 119 we deleted the term “four stages” to reduce redundancy.*

5. Again (see comment B.1), Figure 3c shows the proportion of cardiac myocyte nuclei in fetal (n=3), infantile (n=5), adult non-failing (n=5) and adult failing (n=5) LV tissue given. How can it be that for fetal heart the given number of samples and data points does not fit to the given number of samples in Table S1 and S2 (i.e., 3 versus 8)?

6. Figure 3d shows the distribution of cardiac myocyte ploidy in the different LV tissues. The given number of samples does not fit to the given number of samples in Table S1 and S2 (i.e., fetal 3 versus 8, infantile 6 versus 5, failing 4 versus 5)?

Response to questions 5 & 6: *For the revised version we provide the proportion of cardiac myocyte nuclei and ploidy for all samples listed in Suppl. Table S1 (revised Fig. 1c, d). We apologize for the integration of data of an infantile heart without giving the clinical details of the sample. Since the tissue amount from this patient was too low for further analysis, we excluded this data from the revised version of the manuscript.*

7. In Figure S1 is the plot for one infantile and one non-failing heart labeled with some letter ('h'). Moreover, which samples are plotted? The sample IDs should be provided.

Response: *We corrected the labeling of Suppl. Fig. S1.*

8. In Figure S4 is the legend for S4a and S4b mixed. In addition, the Venn diagram contains the number of genes with ≥ 1 FPKM in fetal, infant, adult non-failing or adult failing cardiac myocytes. Is the FPKM threshold on average over all replicates of a group or in at least one replicate?

Response: *We used mean values over all replicates of each group. We added the information to the legend. We added this information also to other legends analyzing genes with selected expression levels. We corrected the order of the legend.*

9. In Figure S9 is the labeling for stable PMRs not correct (i.e., should be 'b' instead of 'f').

Response: *We corrected labeling of Suppl. Fig. S9.*

Reviewer #2 (Remarks to the Author):

Re: Gilsbach et al., Distinct epigenetic programs regulate cardiac myocyte development and disease in the human heart in vivo

In this revised manuscript, the authors were very responsive to prior the prior reviews. They authors added extensive additional data and analyses. In particular, these revisions went a long way to address the common question raised by all reviewers about data reproducibility and the effect of using different tissue samples.

This is a large body of work that will be highly useful for those interested in epigenetic regulation in development, aging, and disease, particularly within cardiomyocytes.

A few remaining concerns:

1. The authors now provide additional information about the tissue samples. The identifiers of the tissue samples are not consistent so that it is not possible to link the data tracks to the individual tissue sample descriptions.

Response: *To comply with the regulations of the ethics committees and data banks, we have to unassign patient information and genomic sequence information. Therefore, sample names and patient IDs in this manuscript are not linked. The same holds true for the order of the samples listed in Table S1 and S2. This information is now mentioned in the methods section.*

2. Fig. S3 shows the correlation between samples for histone ChIP-seq data.

- are these correlations made genome-wide, or at called peaks?

- the clustering analysis shows that samples do not cluster by group. This is concerning for intra-group heterogeneity being greater than inter-group differences.

Response: *Suppl. Fig. S3 displays genome-wide correlations of genic regions. We decided against peaks, since they do not contain regions with low or no ChIP signal and are thus very likely to bias the correlation. In contrast, genome-wide genic regions span low as well as highly enriched regions of all assessed histone marks. Genome-wide correlations are not expected to show group clustering if different stages of a given cell-type are assessed. Clustering of groups is expected if preselected genes/regions are analyzed (i.e. developmental genes or genes with highly variable signal between samples/groups). The*

high correlation values obtained for all ChIP-seqs nicely illustrates the low experimental bias between samples.

3. The comparison to the heart enhancer compendium of Dickel et al (Fig. S7g) should be presented so that one can see the relative number of regions unique to the predictions of this study or of the Dickel study, rather than only the regions in the intersection.

Response: *We modified the figure as suggested (revised Fig. 7h).*

How do the methylation-predicted enhancers compare to ChromHMM in a comparable analysis?

Response: *We added the suggested Figure S11e.*

4. The discussion states, “a triple antibody labeling strategy was necessary and sufficient to establish a sorting strategy for cardiac myocyte nuclei from pre- and postnatal human hearts.” However, the data seems to suggest that individual labeling with a single antibody such as PLN would be sufficient to sort the cardiac myocyte nuclei.

Response: *We agree and changed the statement to: “Thus, labeling of cardiac nuclei with anti-PLN antibodies enables sorting of cardiac myocyte nuclei from pre- and postnatal human hearts.”*

Reviewer #3 (Remarks to the Author):

The authors have provided a wealth of additional material that allows the readers to interpret the study thoroughly. This includes new data, updated data presentation and methodological information. All of this makes it a solid study which will be useful to the scientific community.

Page 5, line 175:were involved in cardiac muscle...

Response: *We corrected the mistake.*

REVIEWERS' COMMENTS:

Reviewer #2 (Remarks to the Author):

The authors were responsive to the critiques and should be commended on a nicely done, important study.

Response to reviewers comments:

Reviewer #2 (Remarks to the Author):

The authors were responsive to the critiques and should be commended on a nicely done, important study.

Response: Thank you very much for reviewing our manuscript.